# Measurement report: Anthropogenic activities reduction suppresses HONO formation: Direct evidence for Secondary Pollution control

Mingzhu Zhai<sup>1,2</sup>, Shengrui Tong<sup>1</sup>, Wenqian Zhang<sup>3</sup>, Hailiang Zhang<sup>1</sup>, Xin Li<sup>4</sup>, Xiaoqi Wang<sup>5</sup>, Maofa Ge<sup>1</sup>

- State Key Laboratory for Structural Chemistry of Unstable and Stable Species, Beijing National Laboratory for Molecular Sciences (BNLMS), CAS Research/Education Center for Excellence in Molecular Sciences, Institute of Chemistry, Chinese Academy of Sciences, Beijing 100190, China
  - <sup>2</sup>University of Chinese Academy of Sciences, Beijing 100049, China
  - <sup>3</sup>NEAC Key Laboratory of Ecology and Environment in Minority Areas, College of Life and Environmental Sciences,
- 10 Minzu University of China, Beijing, China
  - <sup>4</sup>State Key Joint Laboratory of Environmental Simulation and Pollution Control, College of Environmental Sciences and Engineering, Peking University, Beijing 100871, China
  - <sup>5</sup>Key Laboratory of Beijing on Regional Air Pollution Control, Faculty of Environment Science and Engineering, Beijing University of Technology, Beijing 100124, China
- 15 Correspondence to: Shengrui Tong (tongsr@iccas.ac.cn); Maofa Ge (gemaofa@iccas.ac.cn)

Abstract. Nitrous acid (HONO) is a key precursor of atmospheric hydroxyl radicals (OH) and significantly influences the formation of secondary pollutants, making it essential for understanding and controlling air pollution. While many studies have focused on its formation mechanisms, few have explored the impact of variations in anthropogenic activities on HONO formation. Therefore, we investigated the impact of variations in anthropogenic activities on HONO formation based on comprehensive observations conducted in urban Beijing during autumn and winter of 2022. During clean periods with a 53% drop in Traffic Performance Index, HONO, CO, and NO<sub>2</sub> levels decreased by 2–3 times compared to polluted periods and significantly lower than previously reported wintertime levels in Beijing. Source apportionment revealed that NO<sub>2</sub> heterogeneous reaction on ground was the dominant HONO source across all periods. Vehicle emissions contributed more to HONO during clean periods, suggesting that reduced anthropogenic activities has a stronger influence on secondary HONO formation. pNO<sub>3</sub> photolysis contributed more to HONO during polluted periods, due to higher pNO<sub>3</sub> fractions in PM<sub>2.5</sub> under more polluted conditions. Despite including all known formation pathways in the model, unidentified HONO sources still remain. This is strongly associated with intense solar radiation and high OH concentrations at daytime, as well as elevated NH<sub>3</sub> concentrations at nighttime. Emission reduction simulations further revealed that a 50% NOx reduction during polluted periods could lower HONO by up to 46.3%, directly demonstrating that reducing anthropogenic activities significantly suppresses HONO formation and provides a scientific basis for the development of air pollution control strategies.

## 1 Introduction

The worsening global environmental and public health challenges are largely driven by secondary pollution, primarily associated with haze and tropospheric ozone (O<sub>3</sub>). This widespread issue poses substantial risks to both human health and food security (Huang et al., 2014; Wang et al., 2024; Achebak et al., 2024; Chen et al., 2024b; Li et al., 2019). Therefore, identifying the key processes controlling the formation of haze and O<sub>3</sub> pollution is critical for improving understanding and informing effective environmental management strategies at the global scale.

The hydroxyl (OH) radicals play a dominant role in secondary pollution (Wang et al., 2023; Tan et al., 2024). Previous studies have found that the photolysis of HONO not only serves as the "trigger" for daytime photochemical reactions in the early morning, but also acts as an important, and even primary source of OH radicals throughout the whole day (Kim et al., 2014; Xue et al., 2016). During more severe pollution, the contribution of HONO to primary OH radical was higher (70– 92%) (Xue et al., 2020; Slater et al., 2020; Xuan et al., 2024). Due to its importance in atmospheric chemistry, the formation mechanisms of HONO have garnered widespread attention (Zhang et al., 2019a; Cui et al., 2018). HONO can be directly emitted into the atmosphere from primary sources, including fossil fuel combustion (Kurtenbach et al., 2001; Liao et al., 2021; Zhang et al., 2022b; Zhang et al., 2022c), biomass burning (Gu et al., 2020; Theys et al., 2020), and agricultural soils (Xue et al., 2024; Bao et al., 2022), all of which are closely related to anthropogenic activities. Secondary formation of HONO involves gas-phase reactions (Pagsberg et al., 1997; Slater et al., 2020), heterogeneous processes on surfaces (Chen et al., 2023; Yang et al., 2021; Wall and Harris, 2017; Zhang et al., 2023c), and photochemical processes (Ye et al., 2017; Jiang et al., 2024; Chen et al., 2024a). The precursors of these processes (e.g., NO, NO2, and NH3) also predominantly originate from anthropogenic activities (Yu et al., 2021; Liu et al., 2019a). In summary, anthropogenic activities significantly impact HONO formation. Therefore, understanding how variations in anthropogenic activities affect HONO formations and OH radicals is critical for evaluating the effectiveness of human emission controls and for formulating targeted emission reduction policies.

Policy adjustments and energy structure optimization can significantly affect the characteristics and sources of HONO, as well as its contribution to secondary pollution such as PM<sub>2.5</sub> and O<sub>3</sub>. As the capital of China, Beijing is one of the key areas of air pollution under the supervision of Ministry of Ecology and Environment of the People's Republic of China (Li et al., 2020). Over the past decade, Beijing implemented various measures, including the Clean Air Action Plan in 2013 and the Three-Year Action Plan from 2018 to 2020, and moved many heavy-polluting industries out of Beijing to control industrial pollution (Zhang et al., 2016; Chan and Yao, 2008). Additionally, the control of vehicle emissions and coal combustion in Beijing was one of the key tasks (Zhang et al., 2016). With the implementation of these policies, PM<sub>2.5</sub> concentration decreased rapidly, while O<sub>3</sub> concentration increased year by year in Beijing. Moreover, despite the reduction in nitrogen oxides (NO<sub>x</sub>) emissions, the particulate nitrate (pNO<sub>3</sub>) concentration and its proportion in PM<sub>2.5</sub> increased (Zong et al., 2022). The air pollution control focus shifted from single PM<sub>2.5</sub> control to the simultaneous control of both PM<sub>2.5</sub> and O<sub>3</sub> (Liu et al., 2020; Ye et al., 2023). Stagnant meteorological conditions, such as low wind speeds and high relative humidity, are key

drivers of air pollution in autumn and winter. Previous studies have highlighted how such meteorological conditions facilitate the development of secondary pollution in Beijing, with weak southerly winds often driving pollution from industrial regions (Guo et al., 2014; Zheng et al., 2015). In addition, direct anthropogenic emissions, particularly traffic emissions and coal combustions, also are crucial contributors to wintertime PM<sub>2.5</sub> during haze events in northern Chinese cities (Guo et al., 2013; Shen et al., 2024). Thus, both meteorological stagnations and anthropogenic emissions are the primary drivers of secondary pollution formation in Beijing.

While previous studies have generally focused on HONO source and sink analysis (Xuan et al., 2024; Lin et al., 2022; Jia et al., 2020), limited research has explored the impact of variations in anthropogenic activities on HONO formation and atmospheric oxidation under different pollution conditions. Hereby, we conducted a field observation campaign in urban Beijing from 20 September to 23 December 2022, covering the autumn and winter seasons when O<sub>3</sub> and PM<sub>2.5</sub> pollution frequently occurred. HONO and related pollutants (NO, NO<sub>2</sub>, SO<sub>2</sub>, CO, O<sub>3</sub>, NH<sub>3</sub>, pNO<sub>3</sub>, PM<sub>2.5</sub>, and PM<sub>10</sub>), as well as meteorological parameters, were simultaneously measured. During this campaign, stagnant meteorological conditions predominated, with low wind speeds and southerly winds. Therefore, the variations in concentration primarily reflect changes in local emissions caused by anthropogenic activities. It provided a unique opportunity to identify HONO sources and their potential impact on secondary pollution formation in urban Beijing, which has been rarely studied in the past. In addition, a box model coupled with the Regional Atmospheric Chemistry Mechanism version 2 (RACM2) was used to investigate the chemical budget of HONO under different pollution conditions and to quantify the impact of a variations in anthropogenic activities on HONO formation. In summary, through continuous field observations and model simulations, we provide direct evidence that reducing anthropogenic activities is crucial for controlling wintertime HONO formation in Beijing, providing a direct basis for formulating effective air pollution control strategies.

#### 2 Experimental Methods

#### 85 2.1 Measurement site

The measurement site in this study was on the third floor of NO.2 building at Institute of Chemistry, Chinese Academy of Sciences (ICCAS, 39.99° N, 116.32° E) in Beijing. It is a typical urban site with no obvious industrial or agricultural emission sources. This site is surrounded by dense buildings and main roads, including the North Fourth Ring Road about 500 m to the south of the site and Chengfu Road about 200 m to the north. Therefore, it is heavily influenced by vehicle emissions and anthropogenic activities. More details about the measurement site have been described by previous studies (Hou et al., 2016; Zhang et al., 2023b; Zhang et al., 2019b). The mixing ratios of atmospheric HONO, NH<sub>3</sub>, NO, NO<sub>2</sub> and meteorological parameters were measured at ICCAS. In addition, mixing ratios of other trace gases (including O<sub>3</sub>, CO and SO<sub>2</sub>), PM<sub>2.5</sub> and PM<sub>10</sub> were obtained at the Wanliu monitoring station of the Beijing Environmental Monitoring Station. The Wanliu monitoring station was just about 3 km away from our measurement site in this study, so the atmospheric environment at the two sites was also similar. In addition, photolysis frequencies (jo<sub>1D</sub>, j<sub>NO2</sub>, j<sub>HONO</sub>) were acquired

simultaneously at Peking University (39.99° N, 116.31° E), which was 600 m to the west of the ICCAS site. The quantitative measurements of pNO<sub>3</sub> in NR-PM<sub>1</sub> were conducted using an Aerodyne Aerosol Chemical Speciation Monitor (ACSM) at Beijing University of Technology (BJUT, 39.87° N, 116.48° E). The BJUT site is located between the southeastern 3rd and 4th ring roads of Beijing, representing a typical urban area similar to the ICCAS site. The pNO<sub>3</sub> concentration in PM<sub>2.5</sub> was approximated based on the mass fraction of PM<sub>1</sub> in PM<sub>2.5</sub>. pNO<sub>3</sub> accounted for 30.56 % of PM<sub>2.5</sub> mass, which is consistent with the pNO<sub>3</sub> proportion reported in other studies during the same season in the North China Plain (Xiao et al., 2025; Xu et al., 2019). Therefore, the pNO<sub>3</sub> concentrations obtained in this study are reliable for further analysis.

## 2.2 Measurement instruments

In this study, the mixing ratios of atmospheric HONO was measured by a custom-built HONO analyzer that had been used in previous observations (Zhang et al., 2020; Tong et al., 2016; Hou et al., 2016). The measurement principle of the custom-built HONO analyzer is the same as the commercial long path absorption photometer (LOPAP) (Heland et al., 2001), and the measurement procedure is as follows. HONO is rapidly absorbed by the absorbent solution (0.06 mol L<sup>-1</sup> sulfanilamide in 1 mol L<sup>-1</sup> HCl) in a double-channel glass spiral coil, forming a stable diazonium salt which then reacts with dye solution (0.8 mmol L<sup>-1</sup> N- (1-naphthyl) ethylenediamine-dihydrochloride) to form an azo dye. The azo dye is pumped into the 50 cm liquid waveguide capillary cell (LWCC) and simultaneously detected by the spectrometer (USB2000+, Ocean Optics, USA), the optical signal is then converted into a numerical value and presented on the computer. Finally, the concentration of HONO is calculated from Lambert's law and the difference of signals between the two channels.

The relevant meteorological parameters such as temperature (T), relative humidity (RH), wind speed (WS) and wind direction (WD) were measured by the Vaisala Weather Transmitter (WXT520). The NOx (NO and NO<sub>2</sub>) mixing ratio was measured by a NOx analyzer (Thermo Scientific, Model 42i) with a detection limit of 1 ppb. The chemiluminescence (CL) technique could overestimate NO<sub>2</sub> concentrations due to interference from NOy (Villena et al., 2012; Wu et al., 2022a). Details of NO<sub>2</sub> correction were provided in Text S1 of the Supporting Information. The mixing ratio of NH<sub>3</sub> was measured by an NH<sub>3</sub> analyzer (G2103, Picarro) using Cavity Ring-Down Spectroscopy (CRDS).

#### 2.3 Chemical model

A zero-dimensional photochemical box model (Framework for 0-Dimensional Atmospheric Modeling–F0AM) based on RACM2 that neglects the vertical and horizontal transport processes was used to analyze the HONO budget in this study. The model was constrained by observed chemical species and meteorological parameters, with a time resolution of 1 hour. Additionally, an extra 24-hour lifetime for all species was defined in the F0AM to prevent the accumulation of secondary species to unreasonable levels. This was achieved by setting a first-order dilution rate of 1/(24×60×60) s<sup>-1</sup> and setting the background concentration of all species to zero. For more details on this part of the model, refer to (Wolfe et al., 2016). The model was run from September 20 to December 23 2022, with an additional 2 days spin-up on the first day to allow intermediates to reach a steady state.

## 3 Results and discussion

## 3.1 Overview of measurement

#### 3.1.1 General analysis of HONO and related air pollutants

Figure 1 illustrates the hourly time series of meteorological parameters and chemical species concentrations during 20 September to 23 December 2022. The specific event between 5:00–23:00 local time (LT) of December 12 was not considered in the further analysis, for the detailed reason given in Text S2 in the Supporting Information. Throughout the entire campaign, there was a significant variation of temperature (Temp) and relative humidity (RH) due to the span across autumn and winter. However, wind speed (WS) remained at a low level, with a range of 0.18–1.79 m s<sup>-1</sup>, and the wind direction (WD) was primarily from the southeast and southwest. The meteorological conditions represented typical stagnant conditions that promoted the accumulation of pollutants in Beijing. The HONO concentrations ranged from 0.01–4.79 ppb, with a mean value of 1.21 ± 0.94 ppb. As shown in Table 1, the highest HONO concentration in this study was was comparable to or higher than in several previous studies (Hou et al., 2016; Spataro et al., 2013; Li et al., 2021), though still lower than the highest values reported by (Gu et al. 2022). This was because the stagnant conditions contributed to the accumulation of HONO (Shi et al., 2019). However, compared to previous autumn and winter in Beijing, the PM<sub>2.5</sub> concentration observed in this study was significantly lower, highlighting the effectiveness of air pollution control measures implemented in recent years. General analysis of other pollutants and meteorological parameters were detailed in Text S3 of the Supporting Information.

Figure 1: Hourly time series of meteorological parameters (Temp, RH, WS, jhono) and chemical species (HONO, O<sub>3</sub>, NO, NO<sub>2</sub>, NH<sub>3</sub>, CO, PM<sub>2.5</sub>, PM<sub>10</sub>) concentrations from 20 September to 23 December 2022. The blue, yellow and green shades represent DHP, PEP and CLP, respectively. (DHP: Double-High Pollution Period, characterized by double-high levels of both O<sub>3</sub> and PM<sub>2.5</sub>; PEP: PM<sub>2.5</sub> Episodic-cycle Pollution Period, characterized by periodic cycle of PM<sub>2.5</sub> pollution; CLP: Clean Low Pollution Period, characterized by relatively low pollutant concentrations.) The color bar in the second subfigure represents wind direction (WD) in degrees.

Table 1: Comparison of measured HONO at different sites in Beijing urban area.

155

160

| Sampling periods | HONO (ppb)                                                                                                                                                                                                                          | HONO <sub>max</sub> (ppb)                                                                                                                                                                                                                                                                                                                                                                                                              | Reference              |
|------------------|-------------------------------------------------------------------------------------------------------------------------------------------------------------------------------------------------------------------------------------|----------------------------------------------------------------------------------------------------------------------------------------------------------------------------------------------------------------------------------------------------------------------------------------------------------------------------------------------------------------------------------------------------------------------------------------|------------------------|
| 2019.6.13-7.4    | 0.44±0.24                                                                                                                                                                                                                           | 1.39                                                                                                                                                                                                                                                                                                                                                                                                                                   | (Li et al., 2021)      |
| 2014.2.22–3.2    | 1.84 (Day),                                                                                                                                                                                                                         | 3.24                                                                                                                                                                                                                                                                                                                                                                                                                                   | (Hou et al., 2016)     |
|                  | 2.06 (Night)                                                                                                                                                                                                                        |                                                                                                                                                                                                                                                                                                                                                                                                                                        |                        |
| 2021.3.1–3.30    | $1.48 \pm 1.09$                                                                                                                                                                                                                     | 4.87                                                                                                                                                                                                                                                                                                                                                                                                                                   | (Zhang et al., 2023b)  |
| 2007.1.23-2.14   | $1.04 \pm 0.73$                                                                                                                                                                                                                     | 2.67                                                                                                                                                                                                                                                                                                                                                                                                                                   | (Spataro et al., 2013) |
| 2007.8.2-8.31    | $1.45 \pm 0.58$                                                                                                                                                                                                                     | 2.91                                                                                                                                                                                                                                                                                                                                                                                                                                   |                        |
| 2018.8.18-9.16   | $0.38 \pm 0.35$                                                                                                                                                                                                                     | 1.87                                                                                                                                                                                                                                                                                                                                                                                                                                   | (Xuan et al., 2023)    |
| 2017.12.1-       | 1.38                                                                                                                                                                                                                                | -                                                                                                                                                                                                                                                                                                                                                                                                                                      | (Lian et al., 2022)    |
| 2018.2.28        |                                                                                                                                                                                                                                     |                                                                                                                                                                                                                                                                                                                                                                                                                                        |                        |
| 2017.5.7-5.30    | $1.25 \pm 0.94$                                                                                                                                                                                                                     | 6.69                                                                                                                                                                                                                                                                                                                                                                                                                                   | (Gu et al., 2022)      |
| 2018.1.15-1.30   | $1.04 \pm 1.27$                                                                                                                                                                                                                     | 9.55                                                                                                                                                                                                                                                                                                                                                                                                                                   |                        |
| 2018.5.25-7.15   | $1.27 \pm 0.44$                                                                                                                                                                                                                     | 2.41                                                                                                                                                                                                                                                                                                                                                                                                                                   |                        |
| 2018.11.26-      | 1.13±0.68                                                                                                                                                                                                                           | 3.18                                                                                                                                                                                                                                                                                                                                                                                                                                   | (Liu et al., 2021)     |
| 2019.1.15        |                                                                                                                                                                                                                                     |                                                                                                                                                                                                                                                                                                                                                                                                                                        |                        |
| 2018.10.25-12.7  | $2.52 \pm 1.61$                                                                                                                                                                                                                     | -                                                                                                                                                                                                                                                                                                                                                                                                                                      | (Zhang et al., 2023c)  |
| 2022.9.20-12.23  | $1.21 \pm 0.94$                                                                                                                                                                                                                     | 4.79                                                                                                                                                                                                                                                                                                                                                                                                                                   | This work              |
|                  | 2019.6.13–7.4<br>2014.2.22–3.2<br>2021.3.1–3.30<br>2007.1.23–2.14<br>2007.8.2–8.31<br>2018.8.18–9.16<br>2017.12.1–<br>2018.2.28<br>2017.5.7–5.30<br>2018.1.15–1.30<br>2018.5.25–7.15<br>2018.11.26–<br>2019.1.15<br>2018.10.25–12.7 | $\begin{array}{c} 2019.6.13-7.4 & 0.44\pm0.24 \\ 2014.2.22-3.2 & 1.84 \text{ (Day),} \\ 2.06 \text{ (Night)} \\ 2021.3.1-3.30 & 1.48\pm1.09 \\ 2007.1.23-2.14 & 1.04\pm0.73 \\ 2007.8.2-8.31 & 1.45\pm0.58 \\ 2018.8.18-9.16 & 0.38\pm0.35 \\ 2017.12.1- & 1.38 \\ 2018.2.28 & 1.25\pm0.94 \\ 2018.1.15-1.30 & 1.04\pm1.27 \\ 2018.5.25-7.15 & 1.27\pm0.44 \\ 2018.11.26- & 2019.1.15 \\ 2018.10.25-12.7 & 2.52\pm1.61 \\ \end{array}$ | 2019.6.13-7.4          |

According to the National Ambient Air Quality Standards (NAAQS), during the observation period, there were 6 days O<sub>3</sub> pollution where the daily maximum 8-hour average concentration of O<sub>3</sub> exceeded the Grade II of NAAQS (160 μg m<sup>-3</sup>, equivalent to 82 ppb at 25°C and 1013.25 hPa). Additionally, there were more instances of PM<sub>2.5</sub> pollution days exceeding the Grade II of NAAQS (75 μg m<sup>-3</sup>). These days with O<sub>3</sub> pollution were all concentrated between September 20 and October 2, accompanied by high PM<sub>2.5</sub> concentrations (up to 150 μg m<sup>-3</sup>). This represented a typical scenario of concurrent high levels of O<sub>3</sub> and PM<sub>2.5</sub> pollution frequently observed in cities during autumn (Sun et al., 2009; Qu et al., 2023; Qi et al., 2024), which was associated with meteorological conditions such as relatively high solar radiation intensity, low WS and high RH (Wu et al., 2022b). PM<sub>2.5</sub> pollution were concentrated between October 3 and November 25. This period represented the typical periodic cycle of haze pollution observed in urban China during the autumn and winter, linked to stagnant meteorological conditions, including low air diffusion rates and high RH (Cheng et al., 2018; Zhang et al., 2014; Guo et al., 2014; Zhang et al., 2015). The concentration of PM<sub>2.5</sub> exhibited a typical periodic pattern lasting 4–7 days. Each

cycle began with mass concentrations below 10 μg m<sup>-3</sup>, which then escalated to 100 μg m<sup>-3</sup> within 2–5 days, and even 165 exceeded 150 ug m<sup>-3</sup>. Each PM<sub>2.5</sub> pollution event consistently extended over 2–4 days, surpassing the Grade II of NAAOS (75 μg m<sup>-3</sup>). RH displayed a similar trend to PM<sub>2.5</sub>. Additionally, the concentrations of HONO, CO, NH<sub>3</sub>, and NO<sub>2</sub> progressively increased with the rise in PM<sub>2.5</sub> levels, suggesting a potential connection between these gaseous pollutants and the formation of PM<sub>2.5</sub>, consistent with findings from other studies conducted during the same period in autumn and winter (Slater et al., 2020; Wang et al., 2017; Zhang et al., 2023c). The period from 26 November to 23 December was marked by 170 concentrations of gaseous pollutants (HONO, NO, NO<sub>2</sub>, CO, NH<sub>3</sub>, O<sub>3</sub>) and PM<sub>2.5</sub> which significantly decreased and remained at low levels, the average hourly PM<sub>2.5</sub> concentration and the daily maximum 8-hour average O<sub>3</sub> concentration were less than the Grade I of NAAQS (35 µg m<sup>-3</sup> and 100 µg m<sup>-3</sup>, respectively). In addition, the concentrations of HONO, NO<sub>2</sub>, NH<sub>3</sub>, and CO were 2–3 times lower compared to the previous two periods. Notably, the atmospheric conditions during this period were unfavorable for pollutant dispersion, characterized by low wind speeds (< 2 m s<sup>-1</sup>) and predominantly 175 southerly winds (Figure S3). However, despite these stagnant meteorological conditions, pollutant concentrations significantly decreased.

Therefore, the observation was segmented accordingly based on the concentrations of O<sub>3</sub> and PM<sub>2.5</sub>. The period from September 20 to October 2 was characterized by double-high levels of both O<sub>3</sub> and PM<sub>2.5</sub>, designated as Double-High Pollution Period (DHP, Blue shaded area in Figure 1). The period from October 3 to November 25 was marked by periodic cycle of PM<sub>2.5</sub> pollution, identified as the PM<sub>2.5</sub> Episodic-cycle Pollution Period (PEP, Yellow shaded area in Figure 1). Based on lower pollutant concentrations, the period from 26 November to 23 December was designated as the Clean Low Pollution Period (CLP, Green shaded area in Figure 1). This was markedly different from the widespread haze pollution commonly observed in Beijing during December in other studies (Zhang et al., 2019b; Tong et al., 2016), highlighting the uniqueness of this clean period.

#### 3.1.2 Diurnal variation of chemical species across distinct pollution periods with unique characteristics

Meteorological conditions showed typical seasonal transitions from autumn to winter during the DHP, PEP and CLP, characterized by a gradual decrease in Temp and solar radiation, and a delayed sunrise time. RH exhibited an inverse diurnal pattern compared to Temp, and significantly declined during CLP. Detailed diurnal variations of meteorological parameters are provided in Text S3 and Figures S2–S3 in the Supporting Information. During the DHP, the peak PM<sub>2.5</sub> concentration (60.31 μg m<sup>-3</sup>) appeared at 4:00 LT, then began to decrease, reaching its lowest value (29.08 μg m<sup>-3</sup>) at 17:00 LT, and then accumulated again at night (Figure 2). In contrast, the highest values during the PEP (52.92 μg m<sup>-3</sup>) and the CLP (25.58 μg m<sup>-3</sup>) both appeared at 20:00 LT, gradually decreasing at night and reaching their lowest values (41.06 μg m<sup>-3</sup> and 12.70 μg m<sup>-3</sup>) at 7:00 LT. Influenced by high solar radiation intensity, intense photochemical reactions led to elevated daytime O<sub>3</sub> concentrations during the DHP. At the same time, higher Temp caused a large emission of volatile organic compounds (VOCs) (Qin et al., 2025; Huangfu et al., 2019; Zhang et al., 2024). The high O<sub>3</sub> concentrations promoted secondary organic aerosols (SOA), pNO<sub>3</sub>, pSO<sub>4</sub> and other secondary components through the gas-phase oxidation of VOCs, NOx, and SO<sub>2</sub>, as

well as heterogeneous reactions on particle surfaces, thereby exacerbating PM<sub>2.5</sub> pollution (Huang et al., 2014; Ji et al., 2024; Tao et al., 2024). Additionally, low WS and high RH under stagnant meteorological conditions further intensify the PM<sub>2.5</sub> and O<sub>3</sub> synergistic pollution. The PEP represented typical autumn and winter haze pollution in Beijing, with characteristics similar to those reported in previous haze studies (Shen et al., 2024; Liu et al., 2023). Lower Temp and weaker sunlight reduced photochemical reactions, leading to a significant decrease in O<sub>3</sub> concentrations (Fu et al., 2020), which dropped to one-third of that in the DHP. However, relatively high NO<sub>2</sub> concentrations and high RH promoted the formation of pNO<sub>3</sub>, which was an important component of PM<sub>2.5</sub> (Xu et al., 2019). Meanwhile, low WS resulted in weaker diffusion, further contributing to the recurrence of PM<sub>2.5</sub> pollution (Liu et al., 2023). During the CLP, the relatively small diurnal variation in O<sub>3</sub> concentrations suggested suppressed atmospheric photochemistry and a weaker oxidative capacity during this period. In addition, due to lower gaseous pollutant concentrations and lower RH, PM<sub>2.5</sub> concentration significantly decreased compared to the DHP and PEP. For the diurnal variations of gas pollutants, there were significant differences between the three periods. During the DHP and PEP, traffic-related peaks of NO, NO<sub>2</sub>, CO and NH<sub>3</sub> were generally observed during the morning rush hour (7:00–8:00 LT), then generally remained at lower levels throughout the daytime until concentrations began to rise again during the evening rush hour and built up during the night. But the magnitude and subsequent variations differed by species and periods. For example, NO<sub>2</sub> and NH<sub>3</sub> showed only small enhancements in the PEP, and CO remained elevated after the morning peak during DHP and decrease after 13:00 LT. Notably, the nighttime NO concentration was also very low during the DHP, staying between 0.46–1.17 ppb, which was due to the stronger O<sub>3</sub> consumption of NO (Kurtenbach et al., 2012). During the CLP, the concentrations of NO<sub>2</sub>, CO and NH<sub>3</sub> were lower, and their diurnal variation was not obvious. However, they still exhibited a much weaker peak around 8:00–9:00 LT, which indicated a significant reduction in vehicle emissions. The pollution characteristics of the three periods (DHP, PEP, and CLP) exhibited both distinct differences and certain similarities, forming a sharp vertical contrast while also allowing for a horizontal comparison, meaning cross-study comparisons with published results for the same periods.

Figure 2: The diurnal variations of chemical species (HONO, NO, NO<sub>2</sub>, NH<sub>3</sub>, CO, O<sub>3</sub>, PM<sub>2.5</sub>) and meteorological parameters (Temp, RH) during three periods. The blue, red, and black dotted lines represent DHP, PEP and CLP, respectively.

HONO and NOx (the main precursors of HONO) exhibited similar diurnal variation trends, with higher mixing ratios at night and lower during the day. Specifically, during the DHP and PEP, HONO concentrations significantly peaked at 7:00–8:00 LT (2.71 ppb and 1.86 ppb, respectively) due to nocturnal formation and accumulation as well as direct vehicles emissions in the morning rush hour, then declined due to rapid photolysis in the daytime. Subsequently, due to the absence of photolysis reactions and the arrival of the evening rush hour, the HONO concentration began to accumulate, remaining at a high level throughout the night (1.94–2.33 ppb and 1.67–1.81 ppb, respectively). This phenomenon was also observed in previous studies during PM<sub>2.5</sub> and O<sub>3</sub> pollution in autumn and winter in Beijing (Jia et al., 2020; Li et al., 2021; Liu et al., 2021), which indicated that HONO was a crucial precursor in driving the formation of PM<sub>2.5</sub> and O<sub>3</sub> pollution. Due to the difference in sunrise time and photolysis processes, the peak HONO concentration during the PEP occurred 1 hour later in the morning compared to the DHP. Wang et al. also observed that the decline in HONO concentration varied seasonally with sunrise times, being around 8:00 LT in winter, around 7:00 LT in autumn, and around 6:00 LT in summer and spring (Wang et al., 2017). During the CLP, the HONO concentration significantly decreased and remained at low level (0.28–0.66 ppb). Due to nighttime formation and accumulation, the HONO concentration peaked (0.66 ppb) around midnight, then slowly decreased before sunrise. The HONO concentration did not show a significant increase during the morning rush hour (7:00–8:00 LT), indicating a substantial reduction in vehicle emissions during the CLP. The HONO concentration decreased to its

minimum value (0.28 ppb) at 11:00 LT, then showed an increase around noon, reaching 0.33 ppb at 13:00 LT. Subsequently, HONO began to accumulate again as the sunlight intensity weakened, maintaining levels between 0.55–0.66 ppb throughout the night.

In summary, the DHP and PEP represented common pollution scenarios during the autumn and winter in Beijing, with HONO concentration levels and diurnal variation comparable to those reported in previous studies (Liu et al., 2021; Zhang et al., 2023a; Slater et al., 2020). However, the CLP observed in this study, which occurred during the frequent winter haze pollution period (November 26 to December 23), differed significantly from the pollution situation in the Beijing urban area during the same period in other studies. For example, Tong et al. observed PM<sub>2.5</sub> pollution events (with average PM<sub>2.5</sub> concentrations up to 144 µg m<sup>-3</sup>) accompanied by high HONO concentrations (3 ppb) in urban Beijing from 12 to 22 245 December 2015 (Tong et al., 2016). Zhang et al. captured a severe haze period with PM<sub>2.5</sub> concentrations reaching 418 µg m <sup>3</sup> and HONO concentrations reaching 5.8 ppb at the same site from 16 to 24 December 2016, while a clean period brought by strong winds was also observed (Zhang et al., 2019b). Liu et al. also observed haze pollution in urban Beijing in December 2018 (Liu et al., 2021). Further analysis revealed that the Traffic Performance Index (TPI) in Beijing was relatively low at 250 3.01 in December 2022, significantly lower than usual, indicating reduced vehicle emissions accompanied by a significant decrease in anthropogenic activities. Compared to September, October, and November 2022, the TPI decreased by 53 %, %, and 25 %, respectively. Compared to December 2021, the decrease was as high as 55 % (https://www.bitrc.org.cn/List/index/cid/7.html). These findings demonstrated that even under stagnant regional meteorological conditions, a substantial reduction in direct anthropogenic activities could lead to significant air quality 255 improvements, resulting in cleaner period. Considering the rapid urbanization in China and the sharp increase in vehicle ownership, these findings may also be applicable to other cities or highly industrialized regions, offering valuable insights for formulating scientifically effective emission reduction policies. The following sections will further explore the source contributions of HONO under different pollution scenarios and quantify the specific impact of direct emission reductions on improving PM<sub>2.5</sub> and O<sub>3</sub> pollution.

#### 3.2 HONO budget during nighttime

Vehicle emissions are the most important direct source of HONO in urban areas (Xu et al., 2015; Yun et al., 2017; Yin et al., 2023). As described in diurnal variations and previous studies, CO and NO were major pollutants originating from combustion processes and could be directly emitted into the atmosphere through vehicle emissions in urban areas (Sun et al., 2014; Bond et al., 2013). HONO exhibited a positive correlation with NO or CO, indicating that vehicle emissions played an important role in HONO formation at this observation site (Figure S5). However, it is noteworthy that during the DHP and PEP, which had higher TPI values, the correlation between HONO and NO or CO was lower than that in the CLP. The higher intensity of anthropogenic activities in these periods led to a significant increase in PM<sub>2.5</sub> and NO<sub>2</sub> concentrations, which in turn enhanced the secondary formation of nighttime HONO (e.g., NO<sub>2</sub> heterogeneous reaction on various surfaces), making it the dominant process. In contrast, the lower PM<sub>2.5</sub> concentrations weaken the contribution of secondary HONO

formation during the CLP, resulting in a more pronounced correlation between vehicle emissions and nighttime HONO formation (Jia et al., 2020). To assess the impact of vehicle emissions in this observation period, the local emission factor EF<sub>emis</sub> (=ΔHONO/ΔNOx) was derived based on ambient measurements. As described in Text S4 and Table S2 in the Supporting Information, the minimum EF<sub>emis</sub> (0.0051) in fresh plumes was taken as the upper limit for the direct emission factor to minimize the influence of secondary HONO formation (Su et al., 2008), which was comparable to the value of 0.0051 measured in Beijing (Zhang et al., 2022c), 0.0053 measured in Ji'nan (Li et al., 2018a) and 0.005 measured in Hong Kong (Xu et al., 2015). Since the types of vehicles around this site had not changed, it was reasonable to use the same  $EF_{emis}$ (0.0051) to assess the contribution of vehicle emission to the HONO through the entire observation period. As shown in Figure 3(a), the levels of directly emitted HONO (HONO<sub>emis</sub>) were comparable during the DHP and PEP, while significantly lower during the CLP. HONO<sub>emis</sub> exhibited significant increases during the evening (19:00–20:00 LT) and early morning (05:00–06:00 LT) rush hours in both DHP and PEP, reflecting stronger vehicle emissions compared with CLP. In contrast, no significant increases in HONO<sub>emis</sub> were observed during the CLP, and its nighttime variation remained weak. Although HONO<sub>emis</sub> were relatively high during the DHP and PEP (Figure 3(a)), the contribution of direct emissions was lower than that in the CLP, due to the greater influence of other sources. Vehicle emissions accounted for 8.3 %, 12.5%, and 16.3 % of nighttime HONO during the DHP, PEP, and CLP, respectively, indicating that the relative importance of direct emissions increases under cleaner periods, which was consistent with previous studies (Jia et al., 2020; Zhang et al., 2022c).

Figure 3: The hourly variations of (a)  $HONO_{emis}$  and (b)  $HONO_{emis}$ /HONO at nighttime during three periods. The blue, red, and black dotted lines represent DHP, PEP and CLP, respectively.

The heterogeneous reactions of NO<sub>2</sub> on various surfaces were widely recognized as the primary secondary source of HONO (Spataro and Ianniello, 2014; Kebede et al., 2016; Indarto, 2011). To mitigate the influence of direct emissions, HONO<sub>corr</sub> (HONO-HONO<sub>emis</sub>) was utilized for further analysis. The HONO<sub>corr</sub>/NO<sub>2</sub> ratio was often used to represent the heterogeneous conversion fraction of NO<sub>2</sub> to HONO due to its first-order dependence on NO<sub>2</sub> (Li et al., 2012; Jenkin et al., 1988). The HONO<sub>corr</sub>/NO<sub>2</sub> ratio in DHP, PEP, and CLP were 0.114, 0.086, and 0.085, respectively, which were higher than previous studies (Zhang et al., 2023c; Zhang et al., 2022b; Cui et al., 2018). This indicated that the potential for heterogeneous

conversion from NO<sub>2</sub> to HONO was stronger during this observation period, especially in DHP. The dominant medium for the heterogeneous conversion of NO<sub>2</sub> has been a contentious issue whether aerosol surfaces or ground surfaces are more important. Due to the absence of measurements of aerosol surface density (SA) in this observation period, PM<sub>2.5</sub> concentrations were used as a substitute to determine the impact of aerosols on the conversion of NO<sub>2</sub> to HONO at nighttime (Lu et al., 2018; Cai et al., 2017). The relationship of HONO<sub>corr</sub> with NO<sub>2</sub> and NO<sub>2</sub>×PM<sub>2.5</sub> was demonstrated in Figure S7. HONO<sub>corr</sub> exhibited a significant positive correlation with NO<sub>2</sub>, with correlation coefficients (R<sup>2</sup>) of 0.66, 0.45, and 0.38 300 during the DHP, PEP, and CLP, respectively, indicating that the heterogeneous reaction of NO<sub>2</sub> was an important source of HONO in this observation period. Additionally, the R<sup>2</sup> between HONO<sub>corr</sub> and NO<sub>2</sub>×PM<sub>2.5</sub> was 0.72, 0.48, and 0.35 for the three periods, suggesting that the heterogeneous reaction of NO<sub>2</sub> on aerosol contributed to HONO formation during the DHP and PEP, which was closely associated with the higher PM<sub>2.5</sub> concentrations providing more reactive surfaces. However, compared to previous studies conducted in Beijing during autumn and winter, the correlation between HONOcorr and 305 NO<sub>2</sub>×PM<sub>2.5</sub> during the DHP and PEP was significantly lower (Tong et al., 2015; Nie et al., 2015). For example, (Yan et al., 2015) and (Zhang et al., 2019b) reported that during haze pollution events in Beijing in the mid-2010s, the average PM<sub>2.5</sub> concentration could reach approximately 130 µg m<sup>-3</sup>, with levels during severe haze episodes approaching 311 µg m<sup>-3</sup>. In contrast, the average PM<sub>2.5</sub> concentration during the DHP and PEP in this study was only around 46 µg m<sup>-3</sup>. This discrepancy 310 may be attributed to the implementation of air pollution control policies in recent years, which led to a substantial reduction in PM<sub>2.5</sub> concentrations in Beijing compared to previous years (Huang et al., 2021; Guha et al., 2024), thereby limiting the heterogeneous reactions of NO<sub>2</sub> on aerosols. Particularly during the CLP, PM<sub>2.5</sub> concentrations decreased by 63 % compared to the DHP and PEP, further weakening the role of aerosol in NO<sub>2</sub> heterogeneous reactions to produce HONO. Consequently, compared to NO<sub>2</sub> heterogeneous reactions on ground surfaces, its reaction on aerosol played a more limited role in HONO 315 formation. Notably, HONO<sub>corr</sub> exhibited the strongest correlation with both NO<sub>2</sub> and NO<sub>2</sub>×PM<sub>2.5</sub> during the DHP, likely due to enhanced oxidation of organic and inorganic components, consistent with the high O<sub>3</sub> concentrations observed, which altered the surface reactivity and consequently promoting NO<sub>2</sub> conversion to HONO (George et al., 2015; Ndour et al., 2008). It was observed that the HONO<sub>corr</sub>/NO<sub>2</sub> ratios increased with the increase in RH (<70 %), indicating that the absorbed water on the surface participated in the heterogeneous reactions of NO<sub>2</sub> to HONO (Stutz et al., 2004). Furthermore, once RH 320 exceeded a critical threshold (≥70 %) (Dark blue dots in the Figure S7 (a–b)), the heterogeneous reactions of NO₂ on various surfaces were inhibited, leading to a decrease in HONO<sub>corr</sub>/NO<sub>2</sub> with increasing RH. Similar segmented correlations between HONO<sub>corr</sub> and RH had been observed in previous studies, which had been interpreted as evidence of the nonlinear dependence of NO<sub>2</sub> to HONO conversion efficiency on RH (Qin et al., 2009; Li et al., 2012; Tong et al., 2015). One possible explanation was that the quantity of water layers formed on various surfaces rapidly increased with rising RH, resulting in 325 effective uptake of HONO and making the surfaces less accessible or reducing their reactivity with NO<sub>2</sub> (Yu et al., 2022; Li et al., 2021).

## 3.3 Model simulations for HONO sources and sinks

The base sources of HONO in the model simulation only included the homogeneous reaction of NO with OH ( $S_{NO+OH}$ ), while the sinks of HONO included the dry deposition of HONO ( $L_{dep}$ ), the photolysis of HONO ( $L_{photo}$ ), and the homogeneous reaction of HONO with OH ( $L_{HONO+OH}$ ). The base model could only explain 4.2 %, 19.1 %, and 19.0 % of the observed HONO (HONO<sub>obs</sub>) during the DHP, PEP, and CLP, respectively, which led to an underestimation of OH and  $O_3$  concentrations in the atmosphere (Liu et al., 2019b; Tie et al., 2019). The proportion of soil near the site was minimal and primarily used for landscaping, with almost no fertilization activities. Moreover, the NH<sub>3</sub> concentration was significantly lower than that observed in rural areas of North China where soil emissions were considered (Xue et al., 2021). Therefore, soil emissions were not included in the model. Other known sources of HONO were incorporated into the model simulations, including vehicle direct emissions ( $S_{eims}$ ), the heterogeneous reactions of NO<sub>2</sub> on ground surfaces ( $S_{NO_{2_g}, hv}$ ) and aerosol surfaces ( $S_{NO_{2_g}, hv}$ ), as well as the photolysis of nitrate ( $S_{pNO_3, hv}$ ). The parameterization mechanisms for the production and removal pathways of HONO were detailed in Table 2, and the specific parameter values were presented in Text S6 of the Supporting Information.

Table 2: Parameterized HONO production/loss mechanisms in model simulations.

| Source/<br>Loss              | RACM Mechanisms                                                | Parametrization                                                                                                                                     |
|------------------------------|----------------------------------------------------------------|-----------------------------------------------------------------------------------------------------------------------------------------------------|
| S <sub>emis</sub>            | Direct emission                                                | $EF_{emis}=0.0051$                                                                                                                                  |
| $S_{NO^{+}OH}$               | $NO + OH \rightarrow HONO$                                     | ${ m k}_{ m OH^+NO}$                                                                                                                                |
| $S_{\mathrm{NO}_{2\_g}}$     | $2NO_2 + H_2O \xrightarrow{ground surface} HONO + HNO_3$       | $k_{\text{het-g}} = \frac{1}{8} \times v_{\text{NO}_2} \times \frac{1}{\text{MLH}} \times \gamma_g$                                                 |
| $S_{\mathrm{NO}_{2\_a}}$     | $2NO_2 + H_2O \xrightarrow{aerosol surface} HONO + HNO_3$      | $k_{\text{het-a}} = \frac{1}{8} \times_{V_{NO_2}} \times SA \times \gamma_a$                                                                        |
| $S_{NO_{2\_g,hv}}$           | $2NO_2 + H_2O + hv \xrightarrow{ground surface} HONO + HNO_3$  | $k_{het\text{-}g,hv}\!\!=\!\frac{1}{4}\!\times\!v_{NO_2}\!\times\!\frac{1}{MLH}\!\times\!\gamma_{g,hv}\!\times\!\frac{j_{NO2}}{0.005s^{\text{-}1}}$ |
| $S_{NO_{2\_a,hv}}$           | $2NO_2 + H_2O + hv \xrightarrow{aerosol surface} HONO + HNO_3$ | $k_{\text{het-a,hv}} = \frac{1}{4} \times_{V_{NO_2}} \times SA \times \gamma_{\text{a,hv}} \times \frac{j_{NO2}}{0.005 s^{-1}}$                     |
| $S_{pNO_3,hv}$               | $NO_3^- + hv \rightarrow 0.67HONO + 0.33NO_x$                  | $k_{pNO_3, hv} = EF \times j_{HNO_3}$                                                                                                               |
| $\mathcal{L}_{\text{photo}}$ | $HONO + hv \rightarrow OH + NO$                                | ${ m j}_{ m HONO}$                                                                                                                                  |
| $L_{\rm HONO+OH}$            | $HONO + OH \rightarrow NO_2 + H_2O$                            | $k_{OH+HONO}$                                                                                                                                       |

| $L_{dep}$                | HONO deposition | $k = \frac{v_{HONO}}{BLH}$ |
|--------------------------|-----------------|----------------------------|
| $\mathbf{L}_{	ext{dep}}$ | HONO deposition | $R = \frac{R}{BLH}$        |

As shown in Text S5 of the Supporting Information, the values of  $\gamma_g$  and  $\gamma_{g,hv}$  were set to  $2.94\times10^{-6}$ , while the values of  $\gamma_a$  and  $\gamma_{a,hv}$  were set to  $3.12\times10^{-5}$ . MLH was taken as 50 m in this observation to assess the ground-level sources of HONO (Lee et al., 2016; Xue et al., 2020; Xue et al., 2022). The enhancement factor (EF) was set to 30, a value commonly used in field observations conducted in autumn in Beijing (Zhang et al., 2022a; Xuan et al., 2024). The average dry deposition velocity of HONO ( $v_{HONO}$ ) was taken as 2 cm s<sup>-1</sup> (Harrison et al., 1996).  $v_{NOOO}$ , and  $v_{HONO}$ , were calculated in the RACM mechanisms. BLH represents boundary layer height, with units in meters (m).

After incorporating all HONO sources listed in Table 2, the diurnal variation of simulated HONO (HONO<sub>sim,0</sub>) from different sources was illustrated in Figure 4. The heterogeneous reaction of NO<sub>2</sub> on ground (including  $S_{{\rm NO}_{2\_g}}$  and  $S_{{\rm NO}_{2\_g,\,hv}}$ ) was the main source of HONO during the observations, contributing 45.5 %, 37.8 %, and 44.0 % to HONO<sub>sim.0</sub> in DHP, PEP, and CLP, respectively. The heterogeneous reaction of NO<sub>2</sub> on aerosol surfaces (including  $S_{NO_{2,a,bv}}$ ) contributed 16.7 %, 14.9 %, and 10.5 % to HONO<sub>sim.0</sub> reflecting the variation in PM<sub>2.5</sub> concentrations across the three periods (Table S1). The heterogeneous reactions of NO<sub>2</sub> (on ground and aerosol surfaces) contributed over 50 % of HONO<sub>sim.0</sub>, comparable to previous research, 54.4 % in the Yangtze River Delta region (Shi et al., 2020), and 63 % in urban areas of Beijing (Zhang et al., 2023b). pNO<sub>3</sub> photolysis contributed 12.7 %, 11.7 %, and 5.0 % to HONO<sub>sim.0</sub> during the DHP, PEP, and CLP, respectively. Although traditionally considered minor (Xue et al., 2020; Zhang et al., 2022c), our results suggested that its importance increases under more polluted conditions, consistent with the higher nitrate fractions observed in PM<sub>2.5</sub> composition studies (Zhai et al., 2021; Xu et al., 2019; Xiao et al., 2025). While still lower than the 31-36 % reported in southern China (Fu et al., 2019), pNO<sub>3</sub> photolysis remained a non-negligible source. A sensitivity analysis (Text S6 and Table S3) showed that variations in the EF had a limited effect on HONO formation when EF = 3, but led to a noticeable increase when EF = 300, indicating that EF could influence the contribution of pNO<sub>3</sub> photolysis to HONO production. These results highlight the importance of EF in quantitatively constraining the HONO budget. The homogeneous reaction of NO with OH radicals typically peaked around 8:00–9:00 LT, contributing 8.2 %, 22.5 %, and 21.1 % during the three periods. Vehicle emissions accounted for 17.0 %, 13.0 %, and 19.5 % with a relatively higher contribution during cleaner periods, consistent with previous studies (Jia et al., 2020). Despite including all known sources, a notable portion of HONO<sub>obs</sub> remained unexplained. The unknown HONO concentration (HONO<sub>unknown</sub>=HONO<sub>obs</sub>-HONO<sub>sim.0</sub>) accounted for 50.4 %, 16.9 %, and 7.0 % during the three periods, respectively. Notably, the HONO<sub>unknown</sub>/HONO<sub>obs</sub> increased with pollution severity (higher PM<sub>2.5</sub> and O<sub>3</sub>), suggesting the existence of additional unidentified sources or processes under complex pollution conditions. Overall, while the model captures the major known sources of HONO, the high proportion of unexplained HONO during heavily polluted periods highlights the need for further investigation into missing mechanisms.

Figure 4: The diurnal variations of the HONO<sub>sim.0</sub> from different sources during the DHP, PEP and CLP.

To further investigate the formation mechanisms of HONO<sub>unknown</sub>, this study employed a time-segmented analysis approach. The transitional periods between day and night (i.e., three hours before and after sunrise and sunset) were excluded, then separately analyzing representative daytime (10:00–15:00 LT) and nighttime (22:00–03:00 LT) episodes. This time-segmented analysis approach effectively avoided the interference of diurnal transitions, allowing for a clearer distinction of the characteristics of HONO<sub>unknown</sub> during different episodes. Due to the rapid photolysis of HONO around noontime, the production rate of HONO<sub>unknown</sub> (P<sub>unknown</sub>) during the daytime could be expressed by the following equation:

 $P_{unknown}$  exhibited a positive correlation with OH  $\times$  j<sub>HONO</sub>, with  $R^2$  of 0.69 (Figure S9), suggesting that daytime HONO<sub>unkonwn</sub> might act as a sink for OH radicals. This result was consistent with our previous studies conducted through ground (Lin et al., 2022; Zhang et al., 2023b) and vertical observations (Zhang et al., 2020) in urban Beijing. Through linear regression analysis of nighttime HONO<sub>unknown</sub> with other pollutants and meteorological parameters, significant positive correlations between HONO<sub>unknown</sub> and RH, NO<sub>2</sub>, PM<sub>2.5</sub>, NH<sub>3</sub>, and CO were observed (Figure S10). Given the strong correlations between CO and PM<sub>2.5</sub> (R<sup>2</sup>=0.54) and NH<sub>3</sub> (R<sup>2</sup>=0.59), these pollutants were likely to share common sources, which could explain the observed good correlation between CO and HONO<sub>unknown</sub>. Further multivariate linear regression analysis revealed that, compared to individual pollutants, the composite parameter NO<sub>2</sub>×NH<sub>3</sub>×RH exhibited a significantly stronger correlation with HONO<sub>unknown</sub>. This finding suggested that HONO<sub>unknown</sub> may be related to NH<sub>3</sub>-promoted heterogeneous reactions of NO<sub>2</sub>, with RH playing a crucial role in this process. Li et al. proposed that NH<sub>3</sub> could promote the hydrolysis of NO<sub>2</sub> at the airwater interface to produce HONO, which could be an important pathway for HONO production in NH<sub>3</sub>-rich environments (Li et al., 2018b). Additionally, our group's previous research on haze pollution during the autumn and winter of 2018 demonstrated that the synergistic effect of RH and NH<sub>3</sub> significantly enhanced the NO<sub>2</sub> heterogeneous conversion to HONO, leading to a positive feedback mechanism between haze pollution and HONO production (Zhang et al., 2023c). Notably, RH and NO<sub>2</sub> levels at nighttime in the DHP and PEP were comparable to those observed during haze pollution episodes in the autumn and winter of 2018. Therefore, the enhancement factor (f<sub>NH2,RH</sub>) from Zhang et al.'s work for nighttime HONO

production in the model simulation was applied. Since RH remained consistently below 40 % during the CLP,  $f_{NH_3, RH}$  was set to 1. Additionally, an extra HONO source, enhanced by photolysis and consuming OH radicals, was introduced for daytime HONO production during the three periods. The revised simulation results (HONO<sub>sim,1</sub>) showed good agreement with HONO<sub>obs</sub>, successfully reproducing the HONO variations during the three periods (Figure 5). Due to intermittent light rain, the model exhibited slightly reduced performance during certain periods, such as on September 22 and October 3. These findings have important implications for future studies. First, pathways related to solar radiation and OH radicals should be considered in future studies on daytime HONO sources, as also suggested by recent findings using explainable machine learning approaches (Gao et al., 2024). Second, for typical pollution events characterized by high relative humidity (RH > 40 %) and elevated NH<sub>3</sub> concentrations (> 10 ppb), the synergistic effect of RH and NH<sub>3</sub> in promoting NO<sub>2</sub> heterogeneous reactions at the air-water interface should be fully considered. These insights provide a critical basis for improving parameterization schemes of HONO formation mechanisms in atmospheric chemistry models.

Figure 5: Time series of observed HONO and simulated HONO under different scenarios during the DHP, PEP, and CLP. The black scatter points represent observed HONO (HONO<sub>obs</sub>), the green scatter points represent the revised simulated HONO (HONO<sub>sim,1</sub>), the orange scatter points represent simulated HONO after a 50 % reduction in NOx emissions (HONO<sub>sim (-50 % NOx)</sub>). The data had a temporal resolution of 1 hour.

As mentioned in the "Overview of measurement", the TPI in December decreased by approximately 50 % compared to September and October, reflecting a significant reduction in anthropogenic activities. To directly assess the impact of reduced anthropogenic activities on atmospheric HONO, we assumed a 50 % reduction in NOx emissions during the DHP

and PEP due to decreased anthropogenic activities. This NOx reduction scenario was incorporated as a constraint in the model to obtain simulated HONO concentrations (HONO<sub>sim(-50 % NOx)</sub>). The simulation results (orange scatter points in Figure 5) indicated that NOx reduction led to significant reductions in HONO levels, with declines of 42.7 % and 46.3 % during the DHP and PEP, respectively. These quantitative results confirmed that reduced anthropogenic activities could effectively suppress atmospheric HONO formation, thereby positively contributing to improved air quality. In summary, this study provided the first direct evidence through scenario simulations that reducing anthropogenic activities was a key strategy for controlling atmospheric HONO levels in autumn and winter in Beijing, thereby limiting the formation of secondary pollutants such as PM<sub>2.5</sub> and O<sub>3</sub>. These findings offered important scientific support for formulating effective pollution control strategies.

## **4 Conclusions**

Here, we presented measurements of HONO in urban Beijing during the autumn and winter of 2022, combined with a box model based on RACM2 to analyze its variation patterns and source mechanisms under three typical pollution periods. Particular attention was given to unidentified HONO sources and their implications for atmospheric chemistry.

Our study identified three distinct pollution scenarios: Double-High Pollution period of O<sub>3</sub> and PM<sub>2.5</sub> (DHP), PM<sub>2.5</sub> Episodic-cycle Pollution period (PEP), and Clean Low Pollution period (CLP). Notably, stagnant meteorological conditions predominated, characterized by low wind speeds and southerly winds, suggesting that variations in pollutant concentrations were mainly driven by local emissions associated with anthropogenic activities. The HONO variation characteristics exhibited similarities and differences across the three periods. The average HONO concentrations were 1.71 ppb and 1.46 ppb during the DHP and PEP, respectively, while it decreased more than threefold to 0.50 ppb during the CLP. HONO exhibited similar diurnal variation trends in the three periods, with higher mixing ratios at night and lower during the day. Other pollutants (NO<sub>2</sub>, NH<sub>3</sub>, CO, and PM<sub>2.5</sub>) also showed significantly lower levels (2–3 times) during the CLP compared to the DHP and PEP, while NO concentrations were lowest during the DHP. O<sub>3</sub> concentration was highest during the DHP and were similar during the PEP and CLP. The differences in pollutant concentration were related to the distinct HONO formation mechanisms and conversion frequencies during the three periods, reflecting the variations in atmospheric chemical processes.

The contribution of vehicle emission to nighttime HONO was highest during the CLP. During the DHP and PEP, stronger correlations between nighttime HONO with PM<sub>2.5</sub> and NO<sub>2</sub> indicated a relative greater contribution from heterogeneous reactions, thereby reducing the relative impact of vehicle emission. The NO<sub>2</sub> heterogeneous reaction on ground was the dominant HONO source in all periods, contributing 45.5 %, 37.8 %, and 44.0 % of simulated HONO, respectively. NO<sub>2</sub> heterogeneous reactions on aerosol surfaces contributed 16.7 %, 14.9 %, and 10.5 % in all periods, respectively. pNO<sub>3</sub> photolysis accounted for 12.7 %, 11.7 %, and 5.0 %, consistent with PM<sub>2.5</sub> concentrations in the three periods. The homogeneous reaction of NO with OH contributed 8.2 %, 22.5 %, and 21.1 %, respectively. Notably, although overall

anthropogenic emissions were lower during the CLP, the relative contribution of vehicle emissions to HONO became more significant. Despite incorporating all known sources into the model, significant missing HONO sources remained during the three periods, accounting for 50.4 %, 16.9 %, and 7.0 %, respectively. Correlation analysis indicated that daytime HONO<sub>unknown</sub> was associated with solar radiation and OH radical consumption, while nighttime HONO<sub>unknown</sub> was related to NH<sub>3</sub>-promoted heterogeneous reactions of NO<sub>2</sub>. Including these pathways in the model significantly improved the agreement with observed HONO. Based on TPI and scenario simulations, a 50 % reduction in NOx emissions due to decreased anthropogenic activities during the DHP and PEP would lead to a 42.7 % and 46.3 % decrease in HONO concentrations, respectively. These results had important policy implications for air pollution control. The study indicated that controlling vehicle emissions might be an effective measure to reduce HONO concentrations and improve air quality. However, during haze pollution periods, it is necessary to complement vehicle emission control with integrated multi-source measures, such as reducing NO<sub>2</sub> and NH<sub>3</sub> emissions, to limit the secondary formation of HONO and thereby more effectively reduce air pollution.

In summary, based on comprehensive observations and model simulations of HONO in Beijing during the autumn and winter, this study elucidated the primary source characteristics of HONO under different typical pollution periods and quantitatively assessed the direct impact of variations in anthropogenic activities on its formation. The results highlighted the critical role of controlling anthropogenic emissions in improving air quality and provide strong scientific support for formulating effective air pollution control strategies in urban areas. However, this study focused on urban environments during the autumn and winter, and a systematic understanding of how variations in anthropogenic activities influence HONO in suburban and rural regions remains lacking. Future research should integrate multi-site and multi-seasonal observations with laboratory studies and regional chemical transport modeling to improve our understanding of HONO sources and atmospheric chemistry, and to quantitatively evaluate the impact of variations in anthropogenic activities on air quality across different environments.

#### Data availability

The data presented in this study can be accessed through https://doi.org/10.5281/zenodo.16083849 (Zhai et al., 2025)

## 465 **Supplement**

The supplement will be published alongside this article.

## **Author contribution**

ZMZ contributed to methodology, data curation and analysis, and original draft writing. TSR contributed to conceptualization, investigation, data curation, writing, review and editing, supervision, and funding acquisition. ZWQ and ZHL contributed to data analysis, result discussion, and manuscript commenting. LX and WXQ participated in observational studies and data curation. GMF contributed to conceptualization, investigation, manuscript review, supervision, and funding acquisition.

## **Competing interests**

The contact author has declared that none of the authors has any competing interests.

## 475 Disclaimer

Publisher's note: Copernicus Publications remains neutral with regard to jurisdictional claims made in the text, published maps, institutional affiliations, or any other geographical representation in this paper. While Copernicus Publications makes every effort to include appropriate place names, the final responsibility lies with the authors.

#### **Financial support**

This work was supported by the National Natural Science Foundation of China (Contract No. 42430606, W2521034).

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
