# Peer review of "Measurement report: Anthropogenic activities reduction suppresses HONO formation: Direct evidence for Secondary Pollution control"

_EGUsphere, 2025_

## Author Comment (AC1)

**A point-by-point response to Referee #1**

We sincerely appreciate Referee #1 for the valuable comments and constructive suggestions that helped improve the quality of our manuscript. The following is a point-by-point response to address the referee's comments. The original comments are shown in *black*, and our corresponding responses are presented in *blue*. The new or modified contents in the revised manuscript are marked in *red*.

**Comments from Referee #1:**

This manuscript studied HONO concentrations and its sources in urban Beijing during autumn and winter of 2022. The results showed that NO2 heterogeneous reaction on ground was the dominant HONO source. Vehicle emissions and nitrate photolysis also contributed to HONO concentrations. In general, the research is interesting, the results and discussions are sounds. Here are some technique comments need to be addressed before it can be accepted.

**Response:** Many thanks to Referee #1 for the valuable comments and constructive suggestions, which are significant for improving the quality of the manuscript. We carefully revised and supplemented the manuscript in response to the referee's comments on the technique comments. The following are point-by-point responses to the referee's comments.

**Detailed comments:**

1. L78, studied.

**Response:** Thanks for your valuable comments. Revision has been made as the referee suggested. Lines 80-82 in the revised manuscript:

"It provided a unique opportunity to identify HONO sources and their potential impact to secondary pollution formation in urban Beijing, which has been rarely studied in the past."

2. L114-115, the NO concentrations measured by chemiluminescence NOx analyzer is ok. But the analyzer could overestimate NO2 concentrations due to include other oxidized nitrogen. You can calibrate the data using the method in JGR: Atmospheres, 127, e2021JD036379. https://doi.org/10.1029/2021JD036379.

**Response:** Thanks for your valuable comments. Revision was made as the referee suggested. We corrected CL\_NO2 (Thermo Scientific, Model 42i NOx analyzer) using interference-free CAPS\_NO2 measurements (Teledyne API-N500 NOx analyzer), and provided rigorous field comparison evidence. To improve the precision and accuracy of the NO2 correction, we established separate daytime and nighttime linear regressions. All relevant parameters were recalculated with the corrected NO2, and the model simulations were rerun. The correction methodology and its impacts were described in detail in the Supporting Information (see Text S1 and Figure S1).

Text S1 in the Supporting Information:

"As the most important precursor of HONO, accurate measurement of NO2 was crucial for analyzing HONO formation. A commercial Thermo Scientific analyzer (42i) used in this study could specifically detect NO. The measurement of NO2 was achieved by converting NO2 to NO through a molybdenum converter. However, the chemiluminescence (CL) technique could overestimate NO2 concentrations because of the interference of NOy. These interferences included HONO, HNO3, HNO4, N2O5, NO3, peroxyacetyl nitrate (PANs, RC(O)OONO2), organic nitrates (RONO2), and peroxynitrates (ROONO2) (Villena et al., 2012; Wu et al., 2022). Therefore, the NO2 measured by the CL-NOx analyzer represented the sum of real NO2 and these interfering species. In contrast, the commercial Teledyne API-N500 NOx analyzer was based on cavity attenuated phase shift (CAPS) technique. It could provide direct absorption measurement of NO2 at 450 nm in the blue region of the electromagnetic spectrum, allowed fast and accurate detection of NO2 without interference from water vapor. The only known potential interferences in the typical ambient environment were dicarbonyl compounds such as glyoxal and methylglyoxal, whose concentrations were usually much lower than NO2 mixing ratios (Kebabian et al., 2008). Therefore, NO2 measured by the CAPS-NOx analyzer (CAPS NO2) could be used to correct the NO2 measured by the CL-NOx analyzer (CL NO2).

We conducted a NO2 field campaign at the ICCAS site from September 19 to October 11, 2023, to compare the performance of the CL-NOx and CAPS-NOx analyzers. The sampling inlets of both instruments were placed at the same location, with identical sampling tube lengths, and the analyzers were housed in the same indoor environment to minimize external interference. The results showed that CAPS NO2 and CL NO2 exhibited similar temporal variations (Figure S1(a) and S1(b)). Notably, CL NO2 was consistently higher than CAPS NO2, with a more pronounced difference during the daytime. This discrepancy was mainly attributed to elevated NOy concentrations caused by enhanced photochemical reactions. Consequently, the fraction of CAPS NO2 in CL NO2 displayed a distinct diurnal pattern, being higher at night and lower during the day (Figure S1(c)), which was consistent with previous findings (Xue et al., 2022; Zhang et al., 2022c). Based on this result, we applied separate calibrations for daytime (07:00-18:00 LT) and nighttime (19:00-next 06:00 LT) data. The results indicated strong linear correlations between CAPS NO2 and CL NO2 during both periods ( $R^2 = 0.96$  for daytime and  $R^2 = 0.95$  for nighttime). The regression equations were "y = 0.98x - 2.27" for daytime and "y = 0.99x - 2.29" for nighttime, where y represented CAPS NO2 and x represented CL NO2 (Figure S1(d) and S1(e)). Using these relationships to correct the NO2 data obtained in this study provided a more reasonable estimation of true NO2 concentrations and offered a reliable basis for further analysis."

**Figure S1** Time series (a) and diurnal variations (b) of CAPS\_NO2 and CL\_NO2, the diurnal variations of the fraction of CAPS\_NO2 in CL\_NO2 (c), and scatter plots with linear fits of CAPS\_NO2 versus CL\_NO2 during daytime (d) and nighttime (e).

Additionally, we sincerely appreciate the recommended reference, which provided valuable guidance for the NO2 correction in this study. We added the relevant information and cited the suggested reference in the revised manuscript (Lines 120–122):

"The chemiluminescence (CL) technique could overestimate NO2 concentrations due to interference from NOy (Villena et al., 2012; Wu et al., 2022a). Details of NO2 correction were provided in Text S1 of the Supporting Information."

3. L123-124, delete the Wolfe et al before the bracket. It is the same for other similar references, such as Yan et al. (Yan et al., 2015), Zhang et al. (Zhang et al., 2019b), etc.

**Response:** Thanks for your valuable comments. Revision was made as the referee suggested. Deleted the Wolfe et al., Yan et al., and Zhang et al. before the bracket. Lines 130-131 and lines 318-321 in the revised manuscript:

"For more details on this part of the model, refer to (Wolfe et al., 2016)."

"For example, (Yan et al., 2015) and (Zhang et al., 2019b) reported that during haze pollution events in Beijing in the mid-2010s, the average  $PM_{2.5}$  concentration could reach approximately  $130 \,\mu g \, m^{-3}$ , with levels during severe haze episodes approaching  $311 \,\mu g \, m^{-3}$ ."

4. Figure 1 caption, it is better to define the meaning of DHP, PEP, and CLP. The meaning of the color bar in the 2nd subfigure should also be clarified.

**Response:** Thanks for your valuable comments. Revision was made as the referee suggested. In the caption of Figure 1, the meanings of DHP, PEP, and CLP are clearly defined, and the meaning of the color bar the 2nd subfigure is also specified. Additionally, "WD (°)" is also labeled in the 2nd subfigure. Lines 151-157 in the revised manuscript:

Figure 1: Hourly time series of meteorological parameters (Temp, RH, WS, jHONO) and chemical species (HONO, O3, NO, NO2, NH3, CO, PM2.5, PM10) concentrations from 20 September to 23 December 2022. The blue, yellow and green shades represent DHP, PEP and CLP, respectively. (DHP: Double-High Pollution Period, characterized by double-high levels of both O3 and PM2.5; PEP: PM2.5 Episodic-cycle Pollution Period,

characterized by periodic cycle of PM2.5 pollution; CLP: Clean Low Pollution Period, characterized by relatively low pollutant concentrations.) The color bar in the second subfigure represents wind direction (WD) in degrees.

**5. L163-169, can you shortly explain the reasons why the pollutants concentrations are so low?**

**Response:** Thanks for your valuable comments. During the CLP period (November 26 to December 23), pollutants concentrations were significantly lower, mainly due to a substantial reduction in anthropogenic activities. In urban Beijing, anthropogenic emissions are dominated by vehicle emissions. During the CLP period, the number of vehicles on the traffic arteries near the observation site decreased markedly, and traffic reports also showed a significant decline in the Traffic Performance Index (TPI), which indicates that the reduction in anthropogenic activities was the primary reason for the decrease in pollutants concentrations. The reasons for the reduced pollutants concentrations during the CLP period were also explained in detail in lines 258–264 of the manuscript.

**6. The value used in Table 2 should be listed.**

**Response:** Thanks for your valuable comments. The value used in Table 2 was listed as the referee suggested. Table 2 in the revised manuscript:

| Source/
Loss                             | RACM Mechanisms                                                | Parametrization                                                                                                                                            |
|---------------------------------------------|----------------------------------------------------------------|------------------------------------------------------------------------------------------------------------------------------------------------------------|
| $S_{emis}$                                  | Direct emission                                                | EF emis =0.0051                                                                                                                                 |
| $S_{NO^{+}OH}$                              | $NO + OH \rightarrow HONO$                                     | $k_{\mathrm{OH+NO}}$                                                                                                                                       |
| $S_{NO_{2\_g}}$                             | $2NO_2 + H_2O \xrightarrow{ground surface} HONO + HNO_3$       | $k_{\text{het-g}} = \frac{1}{8} \times_{V_{NO_2}} \times \frac{1}{MLH} \times_{\gamma_g}$                                                                  |
| $S_{NO_{2\_a}}$                             | $2NO_2 + H_2O \xrightarrow{aerosol surface} HONO + HNO_3$      | $k_{\text{het-a}} = \frac{1}{8} \times_{V_{NO_2}} \times SA \times \gamma_a$                                                                               |
| $S_{NO_{2\_g,hv}}$                          | $2NO_2 + H_2O + hv \xrightarrow{ground surface} HONO + HNO_3$  | $k_{\text{het-g,hv}} = \frac{1}{4} \times_{V_{NO_2}} \times \frac{1}{MLH} \times \gamma_{g,\text{hv}} \times \frac{j_{NO2}}{0.005 s^{\text{-1}}}$          |
| $S_{NO_{\underline{2}_{\underline{a}},hv}}$ | $2NO_2 + H_2O + hv \xrightarrow{aerosol surface} HONO + HNO_3$ | $k_{\text{het-a,hv}} = \frac{1}{4} \times v_{\text{NO}_2} \times \text{SA} \times \gamma_{\text{a,hv}} \times \frac{j_{\text{NO}_2}}{0.005 \text{s}^{-1}}$ |
| $S_{pNO_3,hv}$                              | $pNO_3 + hv \rightarrow 0.67HONO + 0.33NO_x$                   | $k_{pNO_3,hv}=EF\times j_{HNO_3}$                                                                                                                          |
| $\mathcal{L}_{\text{photo}}$                | $HONO + hv \rightarrow OH + NO$                                | $\rm j_{HONO}$                                                                                                                                             |
| $L_{\rm HONO+OH}$                           | $HONO + OH \rightarrow NO_2 + H_2O$                            | $k_{\mathrm{OH+HONO}}$                                                                                                                                     |
| $L_{dep}$                                   | HONO deposition                                                | $k = \frac{v_{\text{HONO}}}{BLH}$                                                                                                                          |

As shown in Text S5 of the Supporting Information, the values of  $\gamma_g$  and  $\gamma_{g,hv}$  were set to  $2.94\times10^{-6}$ , while the values of  $\gamma_a$  and  $\gamma_{a,hv}$  were set to  $3.12\times10^{-5}$ . MLH was taken as 50 m in this observation to assess the ground-level sources of HONO (Lee et al., 2016; Xue et al., 2020; Xue et al., 2022). The enhancement factor (EF) was set to 30, a value commonly used in field observations conducted in autumn in Beijing (Zhang et al., 2022a; Xuan et al., 2024). The average dry deposition velocity of HONO ( $v_{HONO}$ ) was taken as 2 cm s-1 (Harrison et al., 1996).  $k_{NO+OH}$ ,  $k_{OH+HONO}$ , and  $j_{HNO3}$  were calculated in the RACM mechanisms. BLH represents boundary layer height, with units in meters (m).

7. L382-385, did you find any relationships between HONO concentrations and solar radiation? If you have, please show the data. Please refer to the publication: Explainable Machine Learning Reveals the Unknown Sources of Atmospheric HONO during COVID-19, ACS EST Air 2024, 1, 1252–1261.

**Response:** Thank you for your valuable comment and for recommending the reference. We analyzed the relationships between HONOunknown, Punknown, and solar radiation. Hourly solar radiation data were derived from the ERA5 reanalysis dataset (ECMWF). As shown in the revised Figure R1 (see below), HONOunknown exhibited a weak positive correlation with solar radiation ( $R^2 = 0.23$ ), while  $P_{unknown}$  showed a moderate positive correlation (R2 = 0.36). These results indicated that both HONOunknown and Punknown were influenced by solar radiation. However, the degree of correlation differs. Punknown represented the production rate of HONO and was directly driven by photochemical processes during daytime, which were enhanced with increasing solar radiation. In contrast, HONOunknown concentrations were determined by a balance between production and removal processes (e.g., photolysis reaction, heterogeneous reaction and homogeneous reaction), which weakened its direct correlation with solar radiation. Therefore, the relatively stronger correlation observed for Punknown supported the hypothesis that unknown HONO sources were photochemically driven during daytime, while the weaker correlation of HONOunknown was consistent with its influence by multiple processes.

**Figure R1.** Correlations between  $HONO_{unknown}$  and  $P_{unknown}$  with solar radiation. The red dashed lines represent linear regression fits.

In addition, we carefully considered your advice and cited the recommended reference (Explainable Machine Learning Reveals the Unknown Sources of Atmospheric HONO during COVID-19, ACS EST Air, 2024, 1, 1252–1261) in the revised manuscript. Lines 411-413 in the revised manuscript:

"First, pathways related to solar radiation and OH radicals should be considered in future studies on daytime HONO sources, as also suggested by recent findings using explainable machine learning approaches (Gao et al., 2024)."

8. The implications of the research should be clarified. Such as in L419-433, the results may indicate that control vehicle emissions could be an effective measures to reduce air pollution, while more measures should be integrated during the haze periods.

**Response:** Thanks for your valuable comments. The implications of the research were clarified as the referee suggested. Lines 467-471 in the revised manuscript:

"These results had important policy implications for air pollution control. The study indicated that controlling vehicle emissions might be an effective measure to reduce HONO concentrations and improve air quality. However, during haze pollution periods, it is necessary to complement vehicle emission control with integrated multi-source measures, such as reducing NO2 and NH3 emissions, to limit the secondary formation of HONO and thereby more effectively reduce air pollution."

**References**

Gao, Z., Wang, Y., Gligorovski, S., Xue, C., Deng, L., Li, R., Duan, Y., Yin, S., Zhang, L., Zhang, Q., and Wu, D.: Explainable Machine Learning Reveals the Unknown Sources of Atmospheric HONO during COVID-19, ACS ES&T Air, 1, 1252-1261, 10.1021/acsestair.4c00087, 2024.

Harrison, R. M., Peak, J. D., and Collins, G. M.: Tropospheric cycle of nitrous acid, J. Geophys. Res. Atmos., 101, 14429-14439, 10.1029/96jd00341, 1996.

Kebabian, P. L., Wood, E. C., Herndon, S. C., and Freedman, A.: A practical alternative to chemiluminescence-based detection of nitrogen dioxide Cavity attenuated phase shift spectroscopy, Environ. Sci. Technol., 42, 6040–6045, 2008.

Lee, J. D., Whalley, L. K., Heard, D. E., Stone, D., Dunmore, R. E., Hamilton, J. F., Young, D. E., Allan, J. D., Laufs, S., and Kleffmann, J.: Detailed budget analysis of HONO in central London reveals a missing daytime source, Atmos. Chem. Phys., 16, 2747-2764, 10.5194/acp-16-2747-2016, 2016.

Villena, G., Bejan, I., Kurtenbach, R., Wiesen, P., and Kleffmann, J.: Interferences of commercial NO2 instruments in the urban atmosphere and in a smog chamber, Atmos. Meas. Tech., 5, 149-159, 10.5194/amt-5-149-2012, 2012.

Wu, D., Zhang, J., Wang, M., An, J., Wang, R., Haider, H., Xu-Ri, Huang, Y., Zhang, Q., Zhou, F., Tian, H., Zhang, X., Deng, L., Pan, Y., Chen, X., Yu, Y., Hu, C., Wang, R., Song, Y., Gao, Z., Wang, Y., Hou, L., and Liu, M.: Global and Regional Patterns of Soil Nitrous Acid Emissions and Their Acceleration of Rural Photochemical Reactions, J. Geophys. Res. Atmos., 127, 10.1029/2021jd036379, 2022a.

Xuan, H., Liu, J., Zhao, Y., Cao, Q., Chen, T., Wang, Y., Liu, Z., Sun, X., Li, H., Zhang, P., Chu, B., Ma, Q., and He, H.: Relative humidity driven nocturnal HONO formation mechanism in autumn haze events of Beijing, npj Clim. Atmos. Sci., 7, 10.1038/s41612-024-00745-8, 2024.

Xue, C., Ye, C., Kleffmann, J., Zhang, W., He, X., Liu, P., Zhang, C., Zhao, X., Liu, C., Ma, Z., Liu, J., Wang, J., Lu, K., Catoire, V., Mellouki, A., and Mu, Y.: Atmospheric measurements at Mt. Tai - Part II: HONO budget and radical (ROx + NO3) chemistry in the lower boundary layer, Atmos. Chem. Phys., 22, 1035-1057, 10.5194/acp-22-1035-2022, 2022.

Xue, C., Zhang, C., Ye, C., Liu, P., Catoire, V., Krysztofiak, G., Chen, H., Ren, Y., Zhao, X., Wang, J., Zhang, F., Zhang, C., Zhang, J., An, J., Wang, T., Chen, J., Kleffmann, J., Mellouki, A., and Mu, Y.: HONO Budget and Its Role in Nitrate Formation in the Rural North China Plain, Environ. Sci. Technol., 54, 11048-11057, 10.1021/acs.est.0c01832, 2020.

Zhang, J., Lian, C., Wang, W., Ge, M., Guo, Y., Ran, H., Zhang, Y., Zheng, F., Fan, X., Yan, C., Daellenbach, K. R., Liu, Y., Kulmala, M., and An, J.: Amplified role of potential HONO sources in O3 formation in North China Plain during autumn haze aggravating processes, Atmos. Chem. Phys., 22, 3275-3302, https://doi.org/10.5194/acp-22-3275-2022, 2022a.

Zhang, X., Tong, S., Jia, C., Zhang, W., Li, J., Wang, W., Sun, Y., Wang, X., Wang, L., Ji, D., Wang, L., Zhao, P., Tang, G., Xin, J., Li, A., and Ge, M.: The Levels and Sources of Nitrous Acid (HONO) in Winter of Beijing and Sanmenxia, J. Geophys. Res. Atmos., 127, 10.1029/2021jd036278, 2022c.

---

## Author Comment (AC2)

**A point-by-point response to Referee #2**

We sincerely appreciate Referee #2 for the valuable comments and constructive suggestions that help improve the quality of our manuscript. The following is a point-by-point response to address the referee's comments. The original comments are shown in *black*, and our corresponding responses are presented in *blue*. The new or modified contents in the revised manuscript are marked in *red*.

**Comments from Referee #2:**

The manuscript uses measurements of gas-phase species and aerosols together with meteorological parameters to separate the measurements into three distinct periods. For each period they use a model to evaluate the sources of HONO and how they vary between the three phases due to changes in anthropogenic emissions. My major concern with the study is the use of a NOx analyzer (Thermo Fisher 42i), which according to the manual uses a molybdenum converter for the NO2 measurements. This is problematic since molybdenum converters are known to overestimate NO2 due to conversion of PANs and other nitrogen containing compounds and NO2 is a key component of the analysis. Additionally, some clarification is required throughout the manuscript, which is commented as minor comments or technical comments to improve the readability of the manuscript.

I recommend the manuscript be published when these concerns are addressed.

**Response:** Many thanks to Referee #2 for the thorough review and valuable comments on our manuscript. We fully understand your major concern regarding the potential overestimation of NO2 when using the NOx analyzer (Thermo Fisher 42i), which is indeed important for the reliability of our study. In the revised manuscript, we will provide a detailed clarification and add corresponding comparison results and analyses. Meanwhile, we will carefully address and revise general comments, major comments, minor comments, and technical comments to further improve the readability and scientific quality of the manuscript. The following are point-by-point responses to the referee's comments.

**General comments:**

I would suggest changing "anthropogenic activities variations" to "variations in anthropogenic activities" throughout the paper as it is easier to read.

**Response:** Thanks for your helpful suggestions. We agree that "variations in anthropogenic activities" is clearer and easier to read. We revised the wording accordingly throughout the manuscript.

**Major comments:**

Line 114-115: Since the NOx analyzer uses a molybdenum converter, it also converts organic nitrates into NO2/NO and potentially also particulate nitrates. Do you somehow take that into account when using the NO2 measurements? How often is the sensitivity of the different channels calibrated? If it isn't taken into account, can you estimate an

**uncertainty on the measurements?**

**Response:** Thanks for your valuable comments. Revision was made as the referee suggested. We corrected CL\_NO2 (Thermo Scientific, Model 42i NOx analyzer) using interference-free CAPS\_NO2 measurements (Teledyne API-N500 NOx analyzer), and provided rigorous field comparison evidence. To improve the precision and accuracy of the NO2 correction, we established separate daytime and nighttime linear regressions. All relevant parameters were recalculated with the corrected NO2, and the model simulations were rerun. The correction methodology and its impacts were described in detail in the Supporting Information (see Text S1 and Figure S1).

**Text S1 in the Supporting Information:**

"As the most important precursor of HONO, accurate measurement of NO2 was crucial for analyzing HONO formation. A commercial Thermo Scientific analyzer (42i) used in this study could specifically detect NO. The measurement of NO2 was achieved by converting NO2 to NO through a molybdenum converter. However, the chemiluminescence (CL) technique could overestimate NO2 concentrations because of the interference of NOy. These interferences included HONO, HNO3, HNO4, N2O5, NO3, peroxyacetyl nitrate (PANs, RC(O)OONO2), organic nitrates (RONO2), and peroxynitrates (ROONO2) (Villena et al., 2012; Wu et al., 2022). Therefore, the NO2 measured by the CL-NOx analyzer represented the sum of real NO2 and these interfering species. In contrast, the commercial Teledyne API-N500 NOx analyzer was based on cavity attenuated phase shift (CAPS) technique. It could provide direct absorption measurement of NO2 at 450 nm in the blue region of the electromagnetic spectrum, allowed fast and accurate detection of NO2 without interference from water vapor. The only known potential interferences in the typical ambient environment were dicarbonyl compounds such as glyoxal and methylglyoxal, whose concentrations were usually much lower than NO2 mixing ratios (Kebabian et al., 2008). Therefore, NO2 measured by the CAPS-NOx analyzer (CAPS NO2) could be used to correct the NO2 measured by the CL-NOx analyzer (CL NO2).

We conducted a NO2 field campaign at the ICCAS site from September 19 to October 11, 2023, to compare the performance of the CL-NOx and CAPS-NOx analyzers. The sampling inlets of both instruments were placed at the same location, with identical sampling tube lengths, and the analyzers were housed in the same indoor environment to minimize external interference. The results showed that CAPS\_NO2 and CL\_NO2 exhibited similar temporal variations (Figure S1(a) and S1(b)). Notably, CL\_NO2 was consistently higher than CAPS\_NO2, with a more pronounced difference during the daytime. This discrepancy was mainly attributed to elevated NOy concentrations caused by enhanced photochemical reactions. Consequently, the fraction of CAPS\_NO2 in CL\_NO2 displayed a distinct diurnal pattern, being higher at night and lower during the day (Figure S1(c)), which was consistent with previous findings (Xue et al., 2022; Zhang et al., 2022c). Based on this result, we applied separate calibrations for daytime (07:00-18:00 LT) and nighttime (19:00-next 06:00 LT) data. The results indicated strong linear correlations between CAPS\_NO2 and CL\_NO2 during both periods (R2 = 0.96 for daytime and R2 = 0.95 for nighttime). The regression

equations were "y = 0.98x - 2.27" for daytime and "y = 0.99x - 2.29" for nighttime, where y represented CAPS\_NO2 and x represented CL\_NO2 (Figure S1(d) and S1(e)). Using these relationships to correct the NO2 data obtained in this study provided a more reasonable estimation of true NO2 concentrations and offered a reliable basis for further analysis."

**Figure S1** Time series (a) and diurnal variations (b) of CAPS\_NO2 and CL\_NO2, the diurnal variations of the fraction of CAPS\_NO2 in CL\_NO2 (c), and scatter plots with linear fits of CAPS\_NO2 versus CL\_NO2 during daytime (d) and nighttime (e).

Additionally, we added the information in the revised manuscript (Lines 120–122):

"The chemiluminescence (CL) technique could overestimate NO2 concentrations due to interference from NOy (Villena et al., 2012; Wu et al., 2022a). Details of NO2

**Minor comments:**

Line 40: Something seems to be missing in the following sentence: "The more severe pollution, and the higher contribution of HONO to primary OH radicals (70–92 %)."

**Response:** Thanks for your helpful suggestions. Revision was made as the referee suggested. Lines 42–43 in the revised manuscript:

"During more severe pollution, the contribution of HONO to primary OH radical was higher (70–92%)."

Line 56-63: The part of the paragraph between "Over the last decade" and "simultaneous control of both PM2.5 and O3" require some grammatical rephrasing. If the climate policies were implemented prior to another past event, then the use of past perfect tense is good, however, when writing "Over the last decade" then it should just be written in past tense. If you add when the air pollution control focus switched in line 62, then that becomes the other event in the past.

**Response:** Thanks to the referee for the clear explanation regarding the use of tense. We agree that the phrase "Over the last decade" should be followed by past tense rather than past perfect tense, unless contrasted with another event in the past. Accordingly, we revised the paragraph in Lines 59–66 to use the past tense consistently.

"Over the past decade, Beijing implemented various measures, including the Clean Air Action Plan in 2013 and the Three-Year Action Plan from 2018 to 2020, and moved many heavy-polluting industries out of Beijing to control industrial pollution (Zhang et al., 2016; Chan and Yao, 2008). Additionally, the control of vehicle emissions and coal combustion in Beijing was one of the key tasks (Zhang et al., 2016). With the implementation of these policies, PM2.5 concentration decreased rapidly, while O3 concentration increased year by year in Beijing. Moreover, despite the reduction in nitrogen oxides (NOx) emissions, the particulate nitrate (pNO3) concentration and its proportion in PM2.5 increased (Zong et al., 2022). The air pollution control focus shifted from single PM2.5 control to the simultaneous control of both PM2.5 and O3 (Liu et al., 2020; Ye et al., 2023)."

Line 61: In the sentence "the nitrate (NO3-) concentration and its proportion in PM2.5 had increased", what do you mean by nitrate? Is it particulate nitrate, organic nitrates, inorganic nitrates, nitrate radicals? I would suggest defining it the first time you use it, since it is used throughout the manuscript.

**Response:** Thanks to the referee for this helpful comment, and we apologize for the lack of clarity in our original wording. Here "nitrate" refers to particulate nitrate (pNO3).

To avoid confusion, we clarified this definition when it first appeared in the manuscript and used pNO3 consistently throughout the manuscript to denote particulate nitrate, as suggested in the second of the "technical comments" given by the referee. Lines 63–65 in the revised manuscript:

"Moreover, despite the reduction in nitrogen oxides (NOx) emissions, the particulate nitrate (pNO3) concentration and its proportion in PM2.5 had increased (Zong et al., 2022)."

Line 136-137: You write "As shown in Table 1, the highest HONO concentration in this study was generally higher than in other studies.", however, 30% of the previous studies have higher maximum HONO concentrations than your study according to Table 1, so maybe rephrase it to represent that.

**Response:** Thanks for your valuable comments. Revision was made as the referee suggested. Lines 143-145 in the revised manuscript:

"As shown in Table 1, the highest HONO concentration in this study was comparable to or higher than in several previous studies, though still lower than the highest values reported by Gu et al. (2022)."

Figure 1: Please define DHP, PEP, and CLP here since the figure is described before the definitions. And describe the colourbar.

**Response:** Thanks for your valuable comments. Revision was made as the referee suggested. In the caption of Figure 1, the meanings of DHP, PEP, and CLP are clearly defined, and the meaning of the color bar the 2nd subfigure is also specified. Additionally, "WD (°)" is also labeled in the 2nd subfigure. Lines 151-157 in the revised manuscript:

Figure 1: Hourly time series of meteorological parameters (Temp, RH, WS, jHONO) and chemical species (HONO, O3, NO, NO2, NH3, CO, PM2.5, PM10) concentrations from 20 September to 23 December 2022. The blue, yellow and green shades represent DHP, PEP and CLP, respectively. (DHP: Double-High Pollution Period, characterized by double-high levels of both O3 and PM2.5; PEP: PM2.5 Episodic-cycle Pollution Period, characterized by periodic cycle of PM2.5 pollution; CLP: Clean Low Pollution Period, characterized by relatively low pollutant concentrations.) The color bar in the second subfigure represents wind direction (WD) in degrees.

Line 195: When you mention NO3- formation, do you mean particulate nitrate? Because if it is particulate nitrate, do you then mean that particulate nitrate formation leads to PM2.5 pollution or does this part of the sentence only refer to the low WS and BLH?

**Response:** Thanks to the referee for pointing this out, and we apologize for the lack of clarity in our original wording. Here "NO3-" indeed refers to particulate nitrate (pNO3). We intended to state that relatively high NO2 concentrations and high RH promoted the formation of pNO3, which is an important component of PM2.5. Meanwhile, low WS resulted in weaker diffusion, further contributing to the recurrence of PM2.5 pollution. Lines 208-211 in the revised manuscript:

"However, relatively high NO2 concentrations and high RH promoted the formation of pNO3 (Xu et al., 2019), which was an important component of PM2.5. Meanwhile, low WS resulted in weaker diffusion, further contributing to the recurrence of PM2.5 pollution (Liu et al., 2023)."

Line 201-203: You write "During the DHP and PEP, NO, NO2, CO and NH3 showed significant peaks during the morning rush hour (7:00–8:00 LT) due to vehicle emissions,

then remained at lower levels throughout the daytime until concentrations began to rise again during the evening rush hour and built up during the night.". This seems like a overgeneralisation since both NO2 and NH3 only show small if any enhancement during the morning rush hour in the PEP phase, CO doesn't reach lower levels after the increase during DHP and as you write in the following sentence NO doesn't increase at nighttime during DHP.

Response: Thanks for your valuable comments, and we apologize for the lack of clarity in our original wording. We agree that our original wording in Lines 201–203 was too general and did not fully capture the differences among species and periods. In the revised manuscript, we rephrased the description to more accurately reflect the observed behaviors. Specifically, compared with the CLP, NO2 indeed exhibited a more obvious morning peak during the PEP, while NH3 showed only a weaker enhancement. CO did not decrease after the morning peak during the DHP. This is because PM2.5 concentrations remained high from 09:00–13:00 LT, and given the strong correlation between CO and PM2.5, CO concentrations also remained elevated, only starting to decrease after 13:00 LT. In addition, NO did not increase at night during DHP due to the stronger O3 consumption of NO, as explained later in the text. To address this, we revised the sentence to avoid overgeneralization and to provide a more accurate description of the variations across species and phases. Lines 215-221 in the revised manuscript:

"During the DHP and PEP, traffic-related peaks of NO, NO2, CO and NH3 were generally observed during the morning rush hour (7:00–8:00 LT), then generally remained at lower levels throughout the daytime until concentrations began to rise again during the evening rush hour and built up during the night. The magnitude and subsequent variations differed by species and periods. For example, NO2 and NH3 showed only small enhancements in the PEP, and CO remained elevated after the morning peak during DHP and decrease after 13:00 LT. Notably, the nighttime NO concentration was also very low during the DHP, staying between 0.46–1.17 ppb, which was due to the stronger O3 consumption of NO (Kurtenbach et al., 2012)."

**Line 205: NO does not look lower during CLP.**

**Response:** Thanks to the referee for pointing this out, and we apologize for the lack of clarity in our original wording. We agree that the statement in the original text was too general. In fact, the decrease of NO during CLP depends on which period it is compared with. When compared with the PEP, which was also characterized by lower O3 levels, NO concentrations during CLP were indeed lower, reflecting reduced vehicle emissions. However, compared with the DHP, where O3 concentrations were relatively high and NO was already suppressed, the NO level during CLP was not significantly lower and even appeared slightly higher. To avoid confusion, we revised the text to more accurately describe these differences. Line 222 in the revised manuscript:

"During the CLP, the concentrations of NO2, CO and NH3 were lower, and their diurnal

Line 209: What do you mean by vertical contrast and horizontal comparison?

**Response:** Thanks to the referee for pointing this out, and we apologize for the lack of clarity in our original wording. By "vertical contrast", we intended to describe the distinct differences in pollution characteristics among the three periods (DHP, PEP, and CLP), that is, the distinct contrast formed among these three periods. By "horizontal comparison", we meant the comparison of our results with those from the same periods reported in other studies. Lines 224-226 in the revised manuscript:

"The pollution characteristics of the three periods (DHP, PEP, and CLP) exhibited both distinct differences and certain similarities, forming a sharp vertical contrast while also allowing for a horizontal comparison, meaning cross-study comparisons with published results for the same periods."

Line 269-270: You write "exhibited significant increases in the evening (~19:00 LT) and early morning (~6:00 LT) during the DHP and PEP", but in Figure 3a it looks like DHP is continuously increasing over the night and PEP is fairly flat.

**Response:** Thanks to the referee for this valuable comment, and we sincerely apologize for the lack of clarity in our original wording. We agree that in Figure 3a, HONOemis shows a continuous increase throughout the night during DHP, while HONOemis appears relatively flat during PEP. Our original statement in Lines 269–270 was not intended to describe the entire nighttime trend, but rather to highlight the specific periods of 19:00–20:00 and 05:00–06:00 LT, which correspond to the evening and morning rush hours and best represent enhanced vehicle emissions. To avoid misunderstanding, we revised the sentence to clarify this focus and to make the comparison with CLP more precise. Lines 288-291 in the revised manuscript:

"HONOemis exhibited significant increases during the evening (19:00–20:00 LT) and early morning (05:00–06:00 LT) rush hours in both DHP and PEP, reflecting stronger vehicle emissions compared with CLP."

Line 305-307: Maybe add that the enhanced oxidation of organic and inorganics during DHP is consistent with the high  $O_3$  concentrations observed.

**Response:** Thanks to the referee for this valuable comment. We agree that the enhanced oxidation of organic and inorganic components during DHP is consistent with the high O3 concentrations observed. To reflect this, we revised the sentence. Lines 327-330 in the revised manuscript:

"Notably, HONOcorr exhibited the strongest correlation with both NO2 and NO2×PM2.5 during the DHP, likely due to enhanced oxidation of organic and inorganic components,

consistent with the high O3 concentrations observed, which altered the surface reactivity and consequently promoted NO2 conversion to HONO (George et al., 2015; Ndour et al., 2008)."

Table 2: The references used for the parameterization should be mentioned.

**Response:** Thanks for your valuable comments. The references used for the parameterization in Table 2 was listed as the referee suggested. Table 2 in the revised manuscript:

| Source/
Loss                             | RACM Mechanisms                                                | Parametrization                                                                                                                                           |
|---------------------------------------------|----------------------------------------------------------------|-----------------------------------------------------------------------------------------------------------------------------------------------------------|
| $S_{emis}$                                  | Direct emission                                                | EF emis =0.0051                                                                                                                                |
| $S_{NO^{+}OH}$                              | $NO + OH \rightarrow HONO$                                     | $k_{OH+NO}$                                                                                                                                               |
| $S_{NO_{2\_g}}$                             | $2NO_2 + H_2O \xrightarrow{ground surface} HONO + HNO_3$       | $k_{\text{het-g}} = \frac{1}{8} \times_{V_{NO_2}} \times \frac{1}{MLH} \times \gamma_g$                                                                   |
| $S_{NO_{2\_a}}$                             | $2NO_2 + H_2O \xrightarrow{aerosol surface} HONO + HNO_3$      | $k_{\text{het-a}} = \frac{1}{8} \times v_{\text{NO}_2} \times SA \times \gamma_a$                                                                         |
| $S_{NO_{2\_g,hv}}$                          | $2NO_2 + H_2O + hv \xrightarrow{ground surface} HONO + HNO_3$  | $k_{het\text{-}g,hv}\!\!=\!\frac{1}{4}\!\times_{V_{NO_2}}\!\!\times\!\frac{1}{MLH}\!\times\!\!\gamma_{g,hv}\!\times\!\frac{j_{NO2}}{0.005s^{\text{-}1}}$  |
| $S_{NO_{\underline{2}_{\underline{a}},hv}}$ | $2NO_2 + H_2O + hv \xrightarrow{aerosol surface} HONO + HNO_3$ | $k_{\text{het-a,hv}} \!\!=\! \frac{1}{4} \!\times_{V_{NO_2}} \!\!\times\! SA \!\!\times\! \gamma_{a,\text{hv}} \!\!\times\! \frac{j_{NO2}}{0.005 s^{-1}}$ |
| $S_{pNO_3,hv}$                              | $pNO_3 + hv \rightarrow 0.67HONO + 0.33NO_x$                   | $k_{pNO_3,hv}\!\!=\!\!EF\!\!\times\!\!j_{HNO_3}$                                                                                                          |
| $\mathcal{L}_{\text{photo}}$                | $HONO + hv \rightarrow OH + NO$                                | $\mathrm{j}_{\mathrm{HONO}}$                                                                                                                              |
| $L_{\text{HONO+OH}}$                        | $HONO + OH \rightarrow NO_2 + H_2O$                            | $k_{\mathrm{OH+HONO}}$                                                                                                                                    |
| $\mathcal{L}_{\text{dep}}$                  | HONO deposition                                                | $k = \frac{v_{\text{HONO}}}{\text{BLH}}$                                                                                                                  |

As shown in Text S5 of the Supporting Information, the values of  $\gamma_g$  and  $\gamma_{g,hv}$  were set to  $2.94\times10^{-6}$ , while the values of  $\gamma_a$  and  $\gamma_{a,hv}$  were set to  $3.12\times10^{-5}$ . MLH was taken as 50 m in this observation to assess the ground-level sources of HONO (Lee et al., 2016; Xue et al., 2020; Xue et al., 2022). The enhancement factor (EF) was set to 30, a value commonly used in field observations conducted in autumn in Beijing (Zhang et al., 2022a; Xuan et al., 2024). The average dry deposition velocity of HONO ( $v_{HONO}$ ) was taken as 2 cm s-1 (Harrison et al., 1996).  $k_{NO+OH}$ ,  $k_{OH+HONO}$ , and  $j_{HNO3}$  were calculated in the RACM mechanisms. BLH represents boundary layer height, with units in meters (m).

Line 340-342: You write in the SI that you use EF=30 for the photolysis of particulate nitrate, however, studies have reported values between 1 and 700 for aerosols (Ye et al., 2016, Romer et al., 2018, Ye et al., 2017) and up to 1700 for urban grime (Baergen and Donaldson, 2013). Recent studies have found that the enhancement factor (EF) for photolysis of particulate nitrate depends on different aerosol parameters and for

example decrease with increasing particulate nitrate (Andersen et al., 2023, Sommariva et al., 2023, Rowlinson et al., 2025). These dependencies are not incorporated in your model and would maybe give a different effect than what you observed (increasing importance of photolysis of particulate nitrate to the HONO formation with increasing particulate nitrate). While it is probably outside the scope of this paper to investigate the impact of different parameterizations of the EF, it would be good with a couple of sentences to discuss these effects and how it might impact your results.

- C. Ye et al., Rapid cycling of reactive nitrogen in the marine boundary layer. Nature **532**, 489–491 (2016).
- C. Ye et al., Photolysis of particulate nitrate as a source of HONO and NOx. Environ. Sci. Technol.**51**, 6849–6856 (2017)
- P. S. Romer et al., Constraints on aerosol nitrate photolysis as a potential source of HONO and NOx. Environ. Sci. Technol.**52**, 13738–13746 (2018).
- A. M. Baergen, D. J. Donaldson, Photochemical renoxification of nitric acid on real urban grime. Environ. Sci. Technol.47, 815–820 (2013).
- S. T. Andersen et al., Extensive field evidence for the release of HONO from the photolysis of nitrate aerosols.Sci. Adv.9, eadd6266(2023)
- R. Sommariva et al., Factors Influencing the Formation of Nitrous Acid from Photolysis of Particulate Nitrate. JPCA 127, 9302-9310 (2023)
- M. J. Rowlinson et al., Observations of tropospheric HONO are incompatible with understanding of atmospheric chemistry, EGUsphere [preprint] (2025)

**Response:** Thanks to the referee for this valuable comment. As you correctly pointed out, the EF for pNO3 photolysis varies widely across different studies and is influenced by various aerosol parameters. Therefore, we added a sensitivity analysis in the Supporting Information to evaluate the impact of EF uncertainty on the contribution of pNO3 photolysis to HONO formation. Text S6 and Table S3 in the Supporting Information:

"EF represented the enhancement factor of the photolysis rate of pNO3 relative to that of HNO3. Laboratory studies reported EF values between 1 and 700 for aerosols (Romer et al., 2018; Ye et al., 2016; Ye et al., 2017), and experimental values up to 1700 for urban grime (Baergen and Donaldson, 2013). However, EF is widely considered to carry substantial uncertainty, which can translate into uncertainty in HONO concentrations. In this study, we adopted a moderate EF (=30) commonly used for autumn in Beijing (Zhang et al., 2022a; Xuan et al., 2024). In addition, to comprehensively evaluate the potential impact of EF uncertainty on the results, a sensitivity analysis was conducted by decreasing and increasing the EF by one order of magnitude (i.e., EF=3 and EF=300). The corresponding changes in HONO concentrations during the three periods were summarized in Table S3. When EF=3, the changes were approximately 3.2 %, 3.4 %, and 2.1 % during DHP, PEP, and CLP, respectively, indicating that the variation in the contribution of pNO3 photolysis to

HONO formation was negligible compared with that under EF = 30. In contrast, when EF=300, the changes were 31.5 %, 34.1 %, and 20.5 %, respectively, suggesting that the contribution of pNO3 photolysis to HONO formation increased slightly relative to the EF=30. These results demonstrated that the EF value could influence the contribution of pNO3 photolysis to HONO formation, highlighting the importance of EF in quantitatively constraining the HONO budget."

**Table S3** Sensitivity study with EF uncertainty for HONO formation processes.

| EF  | DHP    | PEP    | CLP    |
|-----|--------|--------|--------|
| 3   | -3.2 % | -3.4 % | -2.1%  |
| 300 | 31.5 % | 34.1 % | 20.5 % |

Additionally, we added the information in the revised manuscript (Lines 367–370):

"A sensitivity analysis (Text S6 and Table S3) showed that variations in the EF had a limited effect on HONO formation when EF = 3, but led to a noticeable increase when EF = 300, indicating that EF could influence the contribution of pNO3 photolysis to HONO production. These results highlight the importance of EF in quantitatively constraining the HONO budget."

Line 424-425: You write "NO3- photolysis accounted for 12.6 %, 11.8 %, and 4.8 %, consistent with PM2.5 concentrations in three periods, and indicating increasing NO3- fractions in PM2.5 under more polluted conditions.", but is that really what you mean? Since the NO3 is approximated based on the mass fraction of PM1 in PM2.5 (line 100 in the manuscript) and you use the same EF to determine the HONO production for NO3- photolysis is it not just an indication that you have significantly more aerosols available with increasing pollution?

**Response:** Thanks to the referee for this thoughtful comment. Our intention was to emphasize that the enhanced contribution of pNO3 photolysis during more polluted periods was consistent with PM2.5 concentrations. Since the pNO3 concentration was approximated using the PM1/PM2.5 ratio and a constant EF was applied, this trend is more reasonably interpreted as reflecting higher PM2.5 concentrations, rather than a compositional change. In our analysis, the same EF was applied across three periods in order to ensure consistency and comparability of the HONO source budget within this observation study. We are aware that the choice of EF is a challenging issue. Therefore, a sensitivity analysis of the EF was included in the Supporting Information (Text S6 and Table S3) to assess the potential influence of EF uncertainty on these results.

To improve clarity and avoid potential ambiguity, we revised the sentence to focus only on the quantitative contribution of pNO3 photolysis across the three periods and its consistency with  $PM_{2.5}$  concentrations. Lines 456-458 in the revised manuscript:

"pNO3 photolysis accounted for 12.7 %, 11.7 %, and 5.0 %, consistent with PM2.5

**concentrations in the three periods."**

**Technical comments:**

Line 19-20: Change "a comprehensive observation" to "comprehensive observations"

**Response:** Thanks for your valuable comments. Revision was made as the referee suggested. Lines 19-21 in the revised manuscript:

"Therefore, we investigated the impact of variations in anthropogenic activities on HONO formation based on comprehensive observations conducted in urban Beijing during autumn and winter of 2022."

Line 25: I would suggest writing particulate nitrate as pNO3 instead of NO3- to avoid people misunderstanding it for NO3 radicals.

**Response:** Thanks for your valuable comments. We agree that using "pNO3" is clearer and avoids confusion with NO3 radicals. Following the suggestion, we revised "NO3" to "pNO3" throughout the manuscript to avoid misunderstanding.

Line 65: change "development of second pollutions in Beijing" to "development of secondary pollution in Beijing"

**Response:** Thanks for your valuable comments. Revision was made as the referee suggested. Lines 67-69 in the revised manuscript:

"Previous studies have highlighted how such meteorological conditions facilitate the development of secondary pollution in Beijing, with weak southerly winds often driving pollution from industrial regions (Guo et al., 2014; Zheng et al., 2015)."

Line 71: delete "had" in "Hereby, we had conducted"

**Response:** Thanks for your valuable comments. Revision was made as the referee suggested. Lines 75-77 in the revised manuscript:

"Hereby, we conducted a field observation campaign in urban Beijing from 20 September to 23 December 2022, covering the autumn and winter seasons when O3 and PM2.5 pollution frequently occurred."

Line 75: change "During this observations" to either "During these observations" or "During this campaign" or "During this observation period"

**Response:** Thanks for your valuable comments. Revision was made as the referee suggested. Lines 78-79 in the revised manuscript:

"During this campaign, stagnant meteorological conditions predominated, with low wind speeds and southerly winds."

Line 78: replace "to" with "on" in "impact to secondary"

**Response:** Thanks for your valuable comments. Revision was made as the referee suggested. Lines 80-82 in the revised manuscript:

"It provided a unique opportunity to identify HONO sources and their potential impact on secondary pollution formation in urban Beijing, which has been rarely studied in the past."

Line 78: replace "studies" with "studied"

**Response:** Thanks for your valuable comments. Revision was made as the referee suggested. Lines 80-82 in the revised manuscript:

"It provided a unique opportunity to identify HONO sources and their potential impact on secondary pollution formation in urban Beijing, which has been rarely studied in the past."

Line 82: replace "provided" with "provide"

**Response:** Thanks for your valuable comments. Revision was made as the referee suggested. Lines 85-87 in the revised manuscript:

"In summary, through continuous field observations and model simulations, we provide direct evidence that reducing anthropogenic activities is crucial for controlling wintertime HONO formation in Beijing, providing a direct basis for formulating effective air pollution control strategies."

Line 123: remove "(Wolfe et al., 2016)" in "refer to Wolfe et al (Wolfe et al., 2016)"

**Response:** Thanks for your valuable comments. Revision was made as the referee suggested. Lines 130-131 in the revised manuscript:

"For more details on this part of the model, refer to (Wolfe et al., 2016)."

Line 129: replace "illustrated" with "illustrates"

**Response:** Thanks for your valuable comments. Revision was made as the referee suggested. Lines 136-137 in the revised manuscript:

"Figure 1 illustrates the hourly time series of meteorological parameters and chemical

species concentrations during 20 September to 23 December 2022."

Line 132: replace "observation" with "campaign" or "observation period"

**Response:** Thanks for your valuable comments. Revision was made as the referee suggested. Lines 138-140 in the revised manuscript:

"Throughout the entire campaign, there was a significant variation of temperature (Temp) and relative humidity (RH) due to the span across autumn and winter."

Line 132: Add "the" to write "due to the span across"

**Response:** Thanks for your valuable comments. Revision was made as the referee suggested. Lines 138-140 in the revised manuscript:

"Throughout the entire campaign, there was a significant variation of temperature (Temp) and relative humidity (RH) due to the span across autumn and winter."

Line 135: delete "the" in front of Beijing

**Response:** Thanks for your valuable comments. Revision was made as the referee suggested. Lines 141-142 in the revised manuscript:

"The meteorological conditions represented typical stagnant conditions that promoted the accumulation of pollutants in Beijing."

Line 147-148: replace "when" with "where" to write "there were 6 days O3 pollution where the daily maximum"

**Response:** Thanks for your valuable comments. Revision was made as the referee suggested. Lines 159-161 in the revised manuscript:

"According to the National Ambient Air Quality Standards (NAAQS), during the observation period, there were 6 days O3 pollution where the daily maximum 8-hour average concentration of O3 exceeded the Grade II of NAAQS (160 μg m-3, equivalent to 82 ppb at 25°C and 1013.25 hPa)."

Line 151: replace "accompanying" with "accompanied"

**Response:** Thanks for your valuable comments. Revision was made as the referee suggested. Lines 162-163 in the revised manuscript:

"These days with  $O_3$  pollution were all concentrated between September 20 and October 2, accompanied by high  $PM_{2.5}$  concentrations (up to 150  $\mu g$  m-3)."

Line 181: replace "parameter" with "parameters"

**Response:** Thanks for your valuable comments. Revision was made as the referee suggested. Lines 194-195 in the revised manuscript:

"Detailed diurnal variations of meteorological parameters are provided in Text S3 and Figures S2–S3 in the Supporting Information."

Line 200: replace "gases" with "gas" and add "the" to write "between the three periods"

**Response:** Thanks for your valuable comments. Revision was made as the referee suggested. Lines 214-215 in the revised manuscript:

"For the diurnal variations of gas pollutants, there were significant differences between the three periods."

Figure 2 and 3 text: replace "line graphs" with "lines"

**Response:** Thanks for your valuable comments. Revision was made as the referee suggested. Lines 229-230 and lines 299-300 in the revised manuscript:

"Figure 2: The diurnal variations of chemical species (HONO, NO, NO2, NH3, CO, O3 PM2.5) and meteorological parameters (Temp, RH) during three periods. The blue, red, and black dotted lines represent DHP, PEP and CLP, respectively."

"Figure 3: The hourly variations of (a) HONOemis and (b) HONOemis/HONO at nighttime during three periods. The blue, red, and black dotted lines represent DHP, PEP and CLP, respectively."

Line 213 and 414: I would suggest adding "mixing ratios" after higher

**Response:** Thanks for your valuable comments. Revision was made as the referee suggested. Lines 231-232 and lines 444-445 in the revised manuscript:

"HONO and NOx (the main precursors of HONO) exhibited similar diurnal variation trends, with higher mixing ratios at night and lower during the day."

"HONO exhibited similar diurnal variation trends in the three periods, with higher mixing ratios at night and lower during the day."

Line 217, 224, 225, 226 and 227: I would suggest adding "the" in front of HONO concentration

Response: Thanks for your valuable comments. Revisions was made as the referee

suggested. Lines 234-236, line 242, lines 243-244, lines 244-245 and lines 245-246 in the revised manuscript:

"Subsequently, due to the absence of photolysis reactions and the arrival of the evening rush hour, the HONO concentration began to accumulate, remaining at a high level throughout the night (1.94–2.33 ppb and 1.67–1.81 ppb, respectively)."

"During the CLP, the HONO concentration significantly decreased and remained at low level (0.28–0.66 ppb)."

"Due to nighttime formation and accumulation, the HONO concentration peaked (0.66 ppb) around midnight, then slowly decreased before sunrise."

"The HONO concentration did not show a significant increase during the morning rush hour (7:00–8:00 LT), indicating a substantial reduction in vehicle emissions during the CLP."

"The HONO concentration decreased to its minimum value (0.28 ppb) at 11:00 LT, then showed an increase around noon, reaching 0.33 ppb at 13:00 LT."

**Line 233: add "the" before Beijing**

**Response:** Thanks for your valuable comments. Revision was made as the referee suggested. Lines 251-253 in the revised manuscript:

"However, the CLP observed in this study, which occurred during the frequent winter haze pollution period (November 26 to December 23), differed significantly from the pollution situation in the Beijing urban area during the same period in other studies."

**Line 261, 285, 287 and 292: I would add "period" after observation**

**Response:** Thanks for your valuable comments. Revisions was made as the referee suggested. Lines 280-281, lines 306-307, lines 308-311 and lines 312-314 in the revised manuscript:

"To assess the impact of vehicle emissions in this observation period, the local emission factor  $EF_{emis}$  (= $\Delta HONO/\Delta NOx$ ) was derived based on ambient measurements."

"This indicated that the potential for heterogeneous conversion from NO2 to HONO was stronger during this observation period, especially in DHP."

"Due to the absence of measurements of aerosol surface density (SA) in this observation period, PM2.5 concentrations were used as a substitute to determine the impact of aerosols on the conversion of NO2 to HONO at nighttime (Lu et al., 2018; Cai et al., 2017)."

"HONOcorr exhibited a significant positive correlation with NO2, with correlation coefficients (R2) of 0.66, 0.45, and 0.38 during the DHP, PEP, and CLP, respectively,

indicating that the heterogeneous reaction of NO2 was an important source of HONO in this observation period."

Line 275: replace "cleaner period" with "cleaner periods"

**Response:** Thanks for your valuable comments. Revisions was made as the referee suggested. Lines 294-296 in the revised manuscript:

"Vehicle emissions accounted for 9.6 %, 11.7 %, and 17.6 % of nighttime HONO during the DHP, PEP, and CLP, respectively, indicating that the relative importance of direct emissions increases under cleaner periods, which was consistent with previous studies (Jia et al., 2020; Zhang et al., 2022c)."

Line 294: replace "was" with "were"

**Response:** Thanks to the referee for this comment. We carefully checked the sentence in Line 294: "...suggesting that the heterogeneous reaction of NO2 on aerosol contributed to HONO formation during the DHP and PEP, which was closely associated with the higher PM2.5 concentrations providing more reactive surfaces." Here, the antecedent of "which" is "the heterogeneous reaction of NO2 on aerosol," which is singular. Therefore, the use of "was" is grammatically correct.

Line 296-297: remove the double references "Yan et al. (Yan et al., 2015) and Zhang et al. (Zhang et al., 2019b)"

**Response:** Thanks for your valuable comments. Revision was made as the referee suggested. Deleted the Yan et al. and Zhang et al. before the bracket. Lines 318-321 in the revised manuscript:

"For example, (Yan et al., 2015) and (Zhang et al., 2019b) reported that during haze pollution events in Beijing in the mid-2010s, the average  $PM_{2.5}$  concentration could reach approximately  $130 \,\mu g \, m^{-3}$ , with levels during severe haze episodes approaching  $311 \,\mu g \, m^{-3}$ ."

Line 320: replace "further" with "which"

**Response:** Thanks for your valuable comments. Revision was made as the referee suggested. Lines 343-345 in the revised manuscript:

"The base model could only explain 4.2 %, 19.1 %, and 19.0 % of the observed HONO (HONOobs) during the DHP, PEP, and CLP, respectively, which led to an underestimation of OH and O3 concentrations in the atmosphere (Liu et al., 2019b; Tie et al., 2019)."

Line 379, 380, 412, 414, 418, 424, 428: I would add "the" before "three periods"

**Response:** Thanks for your valuable comments. Revisions was made as the referee suggested. Lines 407-408, lines 408-409, lines 442-443, lines 444-445, lines 448-450, lines 456-458, and lines 460-463 in the revised manuscript:

"Additionally, an extra HONO source, enhanced by photolysis and consuming OH radicals, was introduced for daytime HONO production during the three periods."

"The revised simulation results (HONOsim,1) showed good agreement with HONOobs, successfully reproducing the HONO variations during the three periods (Figure 5)."

"The HONO variation characteristics exhibited similarities and differences across the three periods."

"HONO exhibited similar diurnal variation trends in the three periods, with higher mixing ratios at night and lower during the day."

"The differences in pollutant concentration were related to the distinct HONO formation mechanisms and conversion frequencies during the three periods, reflecting the variations in atmospheric chemical processes."

"pNO3 photolysis accounted for 12.7 %, 11.7 %, and 5.0 %, consistent with  $PM_{2.5}$  concentrations in the three periods."

"Despite incorporating all known sources into the model, significant missing HONO sources remained during the three periods, accounting for 50.4 %, 16.9 %, and 7.0 %, respectively."

**Line 397: Replace "declined" with "declines"**

**Response:** Thanks for your valuable comments. Revision was made as the referee suggested. Lines 427-429 in the revised manuscript:

"The simulation results (orange scatter points in Figure 5) indicated that NOx reduction led to significant reductions in HONO levels, with declines of 42.7 % and 46.3 % during the DHP and PEP, respectively."

Line 419: Add "relative" before contribution since the absolute contribute is higher during the other two periods

**Response:** Thanks for your valuable comments. Revision was made as the referee suggested. Lines 451-453 in the revised manuscript:

"During the DHP and PEP, stronger correlations between nighttime HONO with PM2.5 and NO2 indicated a relative greater contribution from heterogeneous reactions, thereby reducing the relative impact of vehicle emission."

**Line 421: Add "the" before dominant**

**Response:** Thanks for your valuable comments. Revision was made as the referee suggested. Lines 453-455 in the revised manuscript:

"The NO2 heterogeneous reaction on ground was the dominant HONO source in all periods, contributing 45.5%, 37.8 %, and 44.0 % of simulated HONO, respectively."

Line 431: "significantly reproduced" should be replaced by "significantly improved the agreement with"

**Response:** Thanks for your valuable comments. Revision was made as the referee suggested. Line 464 in the revised manuscript:

"Including these pathways in the model significantly improved the agreement with observed HONO."

**References**

Andersen, S. T., Carpenter, L. J., Reed, C., Lee, J. D., Chance, R., Sherwen, T., Vaughan, A. R., Stewart, J., Edwards, P. M., Bloss, W. J., Sommariva, R., Crilley, L. R., Nott, G. J., Neves, L., Read, K., Heard, D. E., Seakins, P. W., Whalley, L. K., Boustead, G. A., Fleming, L. T., Stone, D., and Fomba, K. W.: Extensive field evidence for the release of HONO from the photolysis of nitrate aerosols, Sci. Adv., 9, eadd6266, https://doi.org/10.1126/sciadv.add6266, 2023.

Baergen, A. M. and Donaldson, D. J.: Photochemical renoxification of nitric acid on real urban grime, Environ. Sci. Technol., 47, 815-820, https://doi.org/10.1021/es3037862, 2013.

Harrison, R. M., Peak, J. D., and Collins, G. M.: Tropospheric cycle of nitrous acid, J. Geophys. Res. Atmos., 101, 14429-14439, https://doi.org/10.1029/96jd00341, 1996.

Kebabian, P. L., Wood, E. C., Herndon, S. C., and Freedman, A.: A practical alternative to chemiluminescence-based detection of nitrogen dioxide Cavity attenuated phase shift spectroscopy, Environ. Sci. Technol., 42, 6040–6045, https://doi.org/10.1021/es703204j, 2008.

Lee, J. D., Whalley, L. K., Heard, D. E., Stone, D., Dunmore, R. E., Hamilton, J. F., Young, D. E., Allan, J. D., Laufs, S., and Kleffmann, J.: Detailed budget analysis of HONO in central London reveals a missing daytime source, Atmos. Chem. Phys., 16, 2747-2764, https://doi.org/10.5194/acp-16-2747-2016, 2016.

Romer, P. S., Wooldridge, P. J., Crounse, J. D., Kim, M. J., Wennberg, P. O., Dibb, J. E., Scheuer, E., Blake, D. R., Meinardi, S., Brosius, A. L., Thames, A. B., Miller, D. O., Brune, W. H., Hall, S. R., Ryerson, T. B., and Cohen, R. C.: Constraints on Aerosol Nitrate Photolysis as a Potential Source of HONO and NOx, Environ. Sci. Technol., 52, 13738-13746, https://doi.org/10.1021/acs.est.8b03861, 2018.

Sommariva, R., Alam, M. S., Crilley, L. R., Rooney, D. J., Bloss, W. J., Fomba, K. W., Andersen, S. T., and Carpenter, L. J.: Factors Influencing the Formation of Nitrous Acid from Photolysis of Particulate Nitrate, J Phys. Chem. A, 127, 9302-9310, https://doi.org/10.1021/acs.jpca.3c03853, 2023.

Villena, G., Bejan, I., Kurtenbach, R., Wiesen, P., and Kleffmann, J.: Interferences of commercial NO2 instruments in the urban atmosphere and in a smog chamber, Atmos. Meas. Tech., 5, 149-159, https://doi.org/10.5194/amt-5-149-2012, 2012.

Wu, D., Zhang, J., Wang, M., An, J., Wang, R., Haider, H., Xu-Ri, Huang, Y., Zhang, Q., Zhou, F., Tian, H., Zhang, X., Deng, L., Pan, Y., Chen, X., Yu, Y., Hu, C., Wang, R., Song, Y., Gao, Z., Wang, Y., Hou, L., and Liu, M.: Global and Regional Patterns of Soil Nitrous Acid Emissions and Their Acceleration of Rural Photochemical Reactions, J. Geophys. Res. Atmos., 127, https://doi.org/10.1029/2021jd036379, 2022a.

Xuan, H., Liu, J., Zhao, Y., Cao, Q., Chen, T., Wang, Y., Liu, Z., Sun, X., Li, H., Zhang, P., Chu, B., Ma, Q., and He, H.: Relative humidity driven nocturnal HONO formation mechanism in autumn haze events of Beijing, npj Clim. Atmos. Sci., 7, https://doi.org/10.1038/s41612-024-00745-8, 2024.

Xue, C., Ye, C., Kleffmann, J., Zhang, W., He, X., Liu, P., Zhang, C., Zhao, X., Liu, C., Ma, Z., Liu, J., Wang, J., Lu, K., Catoire, V., Mellouki, A., and Mu, Y.: Atmospheric measurements at Mt. Tai – Part II: HONO budget and radical ( $RO_x + NO_3$ ) chemistry in the lower boundary layer, Atmos. Chem. Phys., 22, 1035-1057, https://doi.org/10.5194/acp-22-1035-2022, 2022.

Xue, C., Zhang, C., Ye, C., Liu, P., Catoire, V., Krysztofiak, G., Chen, H., Ren, Y., Zhao, X., Wang, J., Zhang, F., Zhang, C., Zhang, J., An, J., Wang, T., Chen, J., Kleffmann, J., Mellouki, A., and Mu, Y.: HONO Budget and Its Role in Nitrate Formation in the Rural North China Plain, Environ. Sci. Technol., 54, 11048-11057, https://doi.org/10.1021/acs.est.0c01832, 2020.

Ye, C., Heard, D. E., and Whalley, L. K.: Evaluation of Novel Routes for NOx Formation in Remote

Regions, Environ. Sci. Technol., 51, 7442-7449, https://doi.org/10.1021/acs.est.6b06441, 2017.

Ye, C., Zhou, X., Pu, D., Stutz, J., Festa, J., Spolaor, M., Tsai, C., Cantrell, C., Mauldin, R. L., 3rd, Campos, T., Weinheimer, A., Hornbrook, R. S., Apel, E. C., Guenther, A., Kaser, L., Yuan, B., Karl, T., Haggerty, J., Hall, S., Ullmann, K., Smith, J. N., Ortega, J., and Knote, C.: Rapid cycling of reactive nitrogen in the marine boundary layer, Nature, 532, 489-491, https://doi.org/10.1038/nature17195, 2016. Zhang, H., Ren, C., Zhou, X., Tang, K., Liu, Y., Liu, T., Wang, J., Chi, X., Li, M., Li, N., Huang, X., and Ding, A.: Improving HONO Simulations and Evaluating Its Impacts on Secondary Pollution in the Yangtze River Delta Region, China, Geophys. Res. Atmos., 129, https://doi.org/10.1029/2024jd041052, 2024.

Zhang, J., Lian, C., Wang, W., Ge, M., Guo, Y., Ran, H., Zhang, Y., Zheng, F., Fan, X., Yan, C., Daellenbach, K. R., Liu, Y., Kulmala, M., and An, J.: Amplified role of potential HONO sources in O3 formation in North China Plain during autumn haze aggravating processes, Atmos. Chem. Phys., 22, 3275-3302, https://doi.org/10.5194/acp-22-3275-2022, 2022a.

Zhang, X., Tong, S., Jia, C., Zhang, W., Li, J., Wang, W., Sun, Y., Wang, X., Wang, L., Ji, D., Wang, L., Zhao, P., Tang, G., Xin, J., Li, A., and Ge, M.: The Levels and Sources of Nitrous Acid (HONO) in Winter of Beijing and Sanmenxia, J. Geophys. Res. Atmos., 127, https://doi.org/10.1029/2021jd036278, 2022c.

---

## Author Response (AR2)

**A point-by-point reply to the comments**

We sincerely appreciate the editor for insightful and constructive comments, which are helpful for the improvement of the manuscript. We have revised the manuscript carefully according to the editor's comments. The following is a point-by-point reply to address the editor's comments. The original comments are presented in black and our responses are in blue, respectively. The new or modified contents in the revised manuscript are marked in red.

**Comments from Editor:**

**Public justification (visible to the public if the article is accepted and published):**

Thank you for your careful consideration of the referee comments. After careful consideration I have determined that most of the comments are adequately addressed, however, there are two comments that remain unclear. I encourage the authors to further clarify these responses so that future readers will understand the material.

**Response:** We sincerely thank the Editor for handling our manuscript. Regarding the two comments that remain unclear, we carefully revisited our responses and revised the corresponding sections in the manuscript to further clarify these points. The following are point-by-point responses to the Editor's comments.

The first comment that requires further clarification is on line 225 on the track changes version of the manuscript. Specifically, it regards the phrase "sharp vertical contrast while also allowing for a horizontal comparison, meaning cross-study comparisons with published results for the same periods." The meaning of vertical and horizontal are unclear to me. Is this referring to vertical and horizontal gradients within the atmosphere? The end of the sentence seems to imply that, but it is unclear if there are measurements from other locations at the same time period that would provide an appropriate comparison. Please clarify the meaning of this sentence.

**Response:** Thanks to the Editor for pointing out the ambiguity in the terms "vertical" and "horizontal" in this sentence. Our intention was not to describe spatial gradients within the atmosphere, but rather to emphasize that the three periods (DHP, PEP, and CLP) could be compared both internally and with other published studies for the same time periods or seasons in other years. To avoid confusion, we have revised the sentence to make the meaning clearer. Lines 217-221 in the revised manuscript:

"The pollution characteristics of the three periods (DHP, PEP, and CLP) showed distinct differences and certain similarities, enabling comparison of the pollution evolution internally among the three periods within this study. In addition, comparisons with other studies conducted during similar periods or seasons in other years helped to highlight the distinct pollution behavior observed in this campaign."

The second comment relates to the new text on lines 290-291 of the track changes

version. I agree with the referee that I do not see a significant increase in either DHP or in PEP in Figure 3a. DHP steadily increases over night with no apparent step change at the times indicated. PEP appears flat and while there is a small increase at the end, it is unclear if that is truly significant. This point about the emissions requires further justification with data or should be removed from the manuscript.

**Response:** Thanks to the Editor for the constructive comment. To improve clarity, we revised the description of HONOemis variations in Figure 3(a) by describing the three periods separately and providing a clearer explanation of their nighttime patterns. The revised text more accurately reflects the observed differences among the three periods. Lines 282-292 in the revised manuscript:

"As shown in Figure 3(a), the directly emitted HONO (HONOemis) exhibited distinct nighttime patterns among the three periods. HONOemis steadily increased over night in DHP, suggesting continuous accumulation driven by persistent vehicle emissions and reduction in boundary layer height. During the PEP, HONOemis remained relatively stable at nighttime and modest increased during both the evening (~19:00 LT) and early morning (~06:00 LT) rush hours, reflecting enhanced traffic activity. HONOemis during the CLP was markedly lower than in DHP and PEP, remained at low levels over night, indicating weaker vehicle emission."

Additionally, please clarify in the captions of Figures 2 and 3 if these are medians or means. Please consider adding variability indicators (standard deviations or interquartile range) as well. That would help the reader to understand the observed variability and better evaluate statements such as "significant" changes as referenced in my previous comment regarding Figure 3.

**Response:** Thanks to the Editor for this helpful suggestion. In the revised manuscript, we clarified in the captions of Figures 2 and 3 that the curves represented the mean values of the corresponding data. To better illustrate the variability, we also added shaded areas indicating the standard deviations for each period. These changes provided a clearer representation of the observed variability and allowed readers to more accurately evaluate the magnitude and significance of the discussed changes. Lines 222-226 and lines 297-301 in the revised manuscript:

"Figure 2: The diurnal variations of chemical species (HONO, NO, NO2, NH3, CO, O3, PM2.5) and meteorological parameters (Temp, RH) during the three periods. The blue, red, and black dotted lines represent the mean hourly values for DHP, PEP, and CLP, respectively. The shaded areas represent half of the standard deviations ( $\pm 0.5\sigma$ )."

"Figure 3: The hourly variations of (a) HONOemis and (b) HONOemis/HONO at nighttime during three periods. The blue, red, and black dotted lines represent the mean hourly values for DHP, PEP, and CLP, respectively. The shaded areas represent half of the standard deviations ( $\pm 0.5\sigma$ )."